# ReviewScore: Misinformed Peer Review Detection with Large Language Models

## Abstract

Peer review serves as a backbone of academic research, but in most AI conferences, the review quality is degrading as the number of submissions explodes. To reliably detect low-quality reviews, we define *misinformed* review points as either "weaknesses" in a review that contain incorrect premises, or "questions" in a review that can be already answered by the paper. We verify that 15.2% of weaknesses and 26.4% of questions are *misinformed* and introduce ReviewScore indicating if a review point is *misinformed*. To evaluate the factuality of each premise of weaknesses, we propose an automated engine that reconstructs every *explicit* and *implicit premise* from a weakness. We build a human expert-annotated ReviewScore dataset to check the ability of LLMs to automate ReviewScore evaluation. Then, we measure human-model agreements on ReviewScore using eight current state-of-the-art LLMs and verify moderate agreements. We also prove that evaluating premise-level factuality shows significantly higher agreements than evaluating weakness-level factuality. A thorough disagreement analysis further supports a potential of fully automated ReviewScore evaluation. [1]

## 1 Introduction

Peer review is an evaluation of academic work by experts to assess its originality, significance, and validity before publication (Kelly et al., 2014; Taylor & Francis, 2025). In AI conferences, as the number of submissions is exponentially increasing, required number of reviewers is also exploding. As a result, review quality is degraded, which undermines the integrity and reliability of a peer-review system (Stelmakh et al., 2021; Cortes & Lawrence, 2021; Shah, 2022; Kim et al., 2025). Due to the importance of the issue, previous works propose criteria for evaluating review quality. However, we observe a trade-off between applicability to reviews and specificity of rubrics. Goldberg et al. (2025) and Du et al. (2024) proposes criteria that could apply to nearly every review but those are quite vague and subjective. Guo et al. (2023), Sadallah et al. (2025), Purkayastha et al. (2025), and Ou et al. (2025) introduce specific and objective criteria but those target narrow scope of reviews. If we could reliably measure review quality based on specific yet applicable criteria, we could filter out misinformed reviews or give feedback to reviewers before the reviews are released.

To resolve this issue, we introduce two specific yet applicable criteria of a review quality: *unanswerability of questions* and *factuality of weaknesses*. To select the criteria, we recruited a group of human experts and let them independently analyze a small subset of ICLR reviews[2]. Specifically, each of them decomposed a review into *review points*, which are formally defined as follows.

**Definition 1** (Review Point). A review point is a single, self-contained unit of evaluation or inquiry in a review–either a weakness or a question–that stands on its own semantically.

Each human evaluated quality of review points, and they discussed trustworthy criteria to detect low-quality. Based on the discussion, we formally define a *misinformed* review point as follows.

**Definition 2** (Misinformed Review Point). A review point is *misinformed* if and only if

- a question stated in a review can be already answered by the paper, or
- a weakness stated in a review is incorrect or contains incorrect premises regarding the paper.

---

[1] We will make the source code and dataset publicly available.

[2] A group of three graduate students studying AI analyzed reviews of ICLR submissions available in Open-Review. Detailed process of this group work is described in Appendix B.

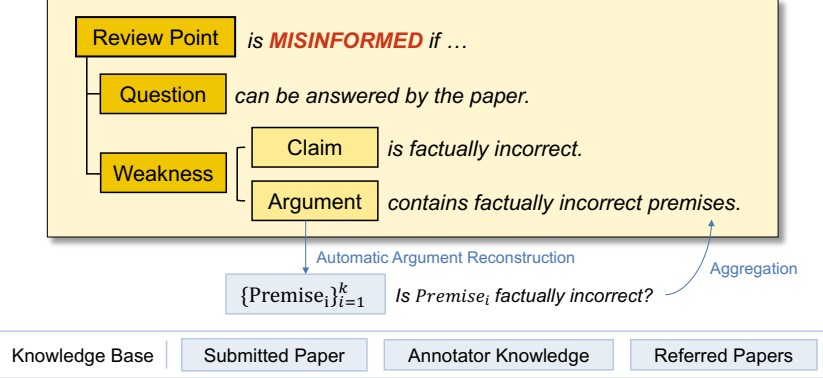

Figure 1: Overview of REVIEWSCORE. Each review point in a review is categorized into question and weakness. We further categorize weakness into claim and argument by the presence of support-ing reasons. Based on an appropriate knowledge base, if a question is answerable by the paper, a claim is factually incorrect, or an argument contains factually incorrect premises, then the review point is *misinformed*. For arguments, to extract all *explicit* and *implicit premises*, we also introduce an automatic argument reconstruction engine.

In this work, the human annotation shows that 26.4% of questions and 15.2% of weaknesses are *misinformed*, which means that the current AI conference reviews contain considerable amount of *misinformed* review points.[3] We also note that these criteria aligns with reviewer guidelines of ma-jor AI conferences. For instance, ACL 2023 Peer Review Policies indicate that "before writing a negative review, check whether your questions are already answered." (Boyd-Graber et al., 2023), and NeurIPS 2025 reviewer guidelines indicate that "reviewers should minimize the chance of mis-understandings during the reviewing process" (NeurIPS 2025 Program Chairs, 2025).

Based on these observations, we define REVIEWSCORE that indicates if a review point is *misin-formed* (Section 3.1). First, we define BASE REVIEWSCORE by directly applying the definition of *misinformed* review points in a 5-point scale. However, we discovered human-annotated factuality often diverges. This is because a weakness often contains both correct and incorrect *explicit premises* or incorrect *implicit premises*, which hinders humans to reliably annotate factuality. To resolve this issue, we further define ADVANCED REVIEWSCORE as an aggregation of premise factuality scores. If a weakness has no supporting reason, we call it as a *claim*, and if a weakness consists of more than one premises, we call it as an *argument*. We adopt two aggregation methods, logical conjunction, following the literature of logic (Beall et al., 2024), and weighted average, maintaining a 5-point scale (Figure 1). Before evaluating ADVANCED REVIEWSCORE, we have to extract all *explicit* and *implicit premises* from an argument. This process is called *argument reconstruction* in logic and critical thinking (Brun & Betz, 2016; Groarke, 2024; Dowden, 2024), and we construct an engine that automatically does this process (Section 3.2). To fully automate *misinformed* review detection with LLMs, we should verify if model-estimated REVIEWSCORE align with human-annotated RE-VIEWSCORE. To do that, we construct a human expert-annotated REVIEWSCORE dataset based on ICLR reviews (Section 3.3). We recruited 15 experienced graduate students and they dedicated total 244 hours for trustworthy human annotation.

We verify the reliability of automatic evaluation of REVIEWSCORE by measuring human-model agreements (Section 4). We use eight current state-of-the-art models, including five proprietary and three open-sourced models, and confirm moderate agreements with the human experts. AD-VANCED REVIEWSCORE clearly outperforms BASE REVIEWSCORE, which proves the effective-ness of premise-level factuality scoring. We also conduct a thorough human-model disagreement analysis and observe that models sometimes misunderstand the meaning of review points or predict scores that minority human annotators give.

To summarize our contributions:

1. We introduce REVIEWSCORE, a novel evaluation criteria that detects *misinformed* review points (i.e., questions that can be answered by the paper or weaknesses with incorrect premises).

2. To evaluate the factuality of premises, we propose an automatic argument reconstruction engine that generates a *valid* and *faithful* set of premises and conclusion.

---

[3]A group of 15 graduate students annotate reviews of 40 works submitted to ICLR 2021-2023. Detailed process of this human annotation is described in Section 3.3.

3. We construct a trustworthy human expert-annotated dataset to measure the reliability of automatic evaluation of REVIEWSCORE.

4. We validate the reliability of automatic REVIEWSCORE evaluation by using eight current state-of-the-art LLMs and conducting a comprehensive human-model disagreement analysis.

## 2 RELATED WORKS

**Peer review evaluation.** Previous works studied how to evaluate the quality of peer reviews. Goldberg et al. (2025) introduces four criteria, *understanding*, *substantiation*, *coverage*, and *constructiveness*. However, they focus on evaluating review as a whole, not each individual review point, which makes difficult to clarify which specific points are *misinformed*. They also do not use LLMs to automate evaluation. Guo et al. (2023) specifically focuses on evaluating *substantiation*. However, they evaluate whether each review point has supporting evidence, not whether the supporting pieces of evidence are factually correct, leaving it as a future work. Our work's *factuality* criteria exactly covers this part. Sadallah et al. (2025) proposes four criteria, *actionability*, *grounding & specificity*, *verifiability*, and *helpfulness*. However, they only focus on evaluating weaknesses, excluding questions, and as in Guo et al. (2023), they only evaluate whether weaknesses are grounded and supported, not whether those are factually correct. Purkayastha et al. (2025) defines *lazy thinking* in the context of NLP paper reviews and introduces heuristic-based rubrics. However, their criteria is only applicable to NLP domain which is hard to be generalized into other domains. Further comparisons with previous works in terms of dataset statistics and review evaluation methods are described in Appendix I.

**Argument evaluation.** In logic and critical thinking, an argument is a list of statements, one of which is the conclusion and the others are the premises (Dutilh Novaes, 2022; Lau & Chan, 2025). To evaluate an argument, we need to follow two steps. First, we have to identify and reconstruct the argument into a set of premises and conclusion, which is called an argument reconstruction (Brun & Betz, 2016; Groarke, 2024; Dowden, 2024). Then, we evaluate whether each premise is factually correct. An argument reconstruction should both be *valid*, which means premises deductively imply a conclusion, and *faithful*, which means premises and a conclusion accurately and completely represents an original argument (Brun & Betz, 2016; Betz & Richardson, 2021). Previously, Betz & Richardson (2021) trains a T5 model for argument reconstruction. However, the training datasets are either synthetic or polished and the reconstruction do not require any additional context information. In contrast, our work targets peer reviews, which include real-world unpolished arguments, and the reconstruction requires an entire paper to fully understand the context of arguments.

## 3 REVIEWSCORE

We newly define REVIEWSCORE that measures how *misinformed* a review point is (Section 3.1). To evaluate REVIEWSCORE, we also introduce an engine that automatically extracts every *explicit* and *implicit premises* from a weakness (Section 3.2). Lastly, we construct a human expert-annotated dataset that evaluates LLM's ability to evaluate REVIEWSCORE (Section 3.3).

### 3.1 DEFINITION

Our goal of defining REVIEWSCORE is to detect *misinformed* review points. Following this goal and the review quality criteria discussion in Section 1, we first define BASE REVIEWSCORE as *factuality of weaknesses* and *unanswerability of questions*. The following definition formally describes it.

**Definition 3** (BASE REVIEWSCORE). Let $x$ be a review point (either a weakness or a question) about a submitted paper $S$. Define

$$\text{Factuality}_S : \mathcal{W} \to \{1, 2, 3, 4, 5\}, \qquad \text{Unanswerability}_S : \mathcal{Q} \to \{1, 2, 3, 4, 5\},$$

where $\mathcal{W}$ and $\mathcal{Q}$ are, respectively, the sets of weaknesses and questions appearing in a review of $S$[4]. We considered score 1–2 as *Misinformed* and score 3–5 as *Not misinformed* for binary classification

---

[4]Detailed rubric is described in Appendix H.

setup. The BASE REVIEWSCORE of $x$ is:

$$\text{ReviewScore}_{\text{base}}(x) = \begin{cases} \text{WScore}(x) = \text{Factuality}_S(x), & \text{if } x \in \mathcal{P} \text{ (weakness)}, \\ \text{QScore}(x) = \text{Unanswerability}_S(x), & \text{if } x \in \mathcal{Q} \text{ (question)}. \end{cases}$$

However, during the group discussion in Section 1, we discovered that human annotators' evaluations on factuality diverge if a weakness contains both factual and nonfactual premises or a nonfactual premise is implicitly presumed. It happens since the human annotators implicitly weigh the importance of underlying premises of a weakness and then decide the final factuality score.

To resolve this issue, we further define ADVANCED REVIEWSCORE. We categorize weaknesses into arguments and claims based on whether there are supporting reasons or not. We keep the definition of WScore to evaluate claims, but further develops a finer-grained score to evaluate arguments. Following the literature of critical thinking (Brun & Betz, 2016; Groarke, 2024; Dowden, 2024), we reconstruct an argument into a premise-conclusion structure and then define ADVANCED REVIEWSCORE for arguments as (an aggregation of) *factuality of premises*. The following definition formally describes it.

**Definition 4** (ADVANCED REVIEWSCORE). Let $x$ be a review point about a submitted paper $S$. Let $\mathcal{C}$, $\mathcal{A}$, and $\mathcal{Q}$ denote, respectively, the sets of claims, arguments, and questions in a review of $S$. For $x \in \mathcal{A}$, let its (*explicit* and *implicit*) premises be $\mathcal{P}(x) = \{p_1, \ldots, p_k\}$ with $\{p_1, \ldots, p_k\} \vdash C$ for the conclusion $C$ of $x$. Let $\mathcal{K}$ be the set of knowledge bases available for factuality judgments (i.e., $S$, annotator knowledge, referred papers), and let

$$\text{Factuality}: \mathcal{U} \times \mathcal{K} \to \{1, 2, 3, 4, 5\}, \qquad \mathcal{U} := \mathcal{C} \cup \bigcup_{x \in \mathcal{A}} \mathcal{P}(x),$$

be a 5-point scoring function for claims and premises given a knowledge base. Define a selector $\text{KB}: \mathcal{U} \to \mathcal{K}$ that chooses the knowledge base used for each item (for claims $x \in \mathcal{C}$, $\text{KB}(x) = S$; for premises $p_i$, $\text{KB}(p_i) = \text{KB}_i \in \mathcal{K}$). For a given $\text{KB}(x) \in \mathcal{K}$, we notate the factuality function as $\text{Factuality}_{\text{KB}(x)}(\cdot)$. Agg is an operator that aggregates a list of scores to a single score. The ADVANCED REVIEWSCORE of $x$ is:

$$\text{ReviewScore}_{\text{adv}}(x) = \begin{cases} \text{ClaimScore}(x) = \text{Factuality}_S(x), & \text{if } x \in \mathcal{C} \text{ (claim)}, \\ \text{ArgScore}(x) = \text{Agg}\big(\text{Factuality}_{\text{KB}_i}(p_i)_{p_i \in \mathcal{P}(x)}\big), & \text{if } x \in \mathcal{A} \text{ (argument)}, \\ \text{QScore}(x) = \text{Unanswerability}_S(x), & \text{if } x \in \mathcal{Q} \text{ (question)}. \end{cases}$$

**Aggregation methods.** To aggregate premise factuality scores as a single ArgScore, we introduce two aggregation methods: logical conjunction and weighted average. Following the literature of logic, an argument is *true* if and only if all premises are *true*. We define a premise is *true* if and only if it has factuality score 3–5, and otherwise, it is *false*. We dubbed this aggregation as logical conjunction, which follows the binary classification setup. However, if an annotator mistakenly evaluates one of premises, then the error propagates to the entire argument. To alleviate this issue, we also aggregate by a weighted average. Since it is difficult to measure the importance of premises, we instead weighted scores by untrivialness of premises (with a scale of 0–2). This is intended to simply filter out trivially true premises by measuring their importance as 0.[5]

## 3.2 AUTOMATIC ARGUMENT RECONSTRUCTION

To evaluate ArgScore, we have to extract (*explicit* and *implicit*) premises $\mathcal{P}(x)$ from an argument $x \in \mathcal{A}$. Since human experts require significant amount of time and costs to do this, we automate it using LLMs. First, we check if a model could directly reconstruct an argument by giving detailed instructions. To preserve the context of an argument, we also give the model a submitted paper $S$. However, it mostly fails to generate *valid* and *faithful* reconstructions[6]. To resolve this issue, we add two feedback loops to ensure *validity* and *faithfulness* of an argument reconstruction (Figure 2a).

---

[5]Since we reconstruct every argument as valid (i.e., a set of premises deductively implies a conclusion), there are often conditional premises that make an argument valid but are trivially true. Detailed rubric of untrivialness is described in Appendix H.

[6]Detailed numerical results and qualitative analysis are reported in Appendix C.

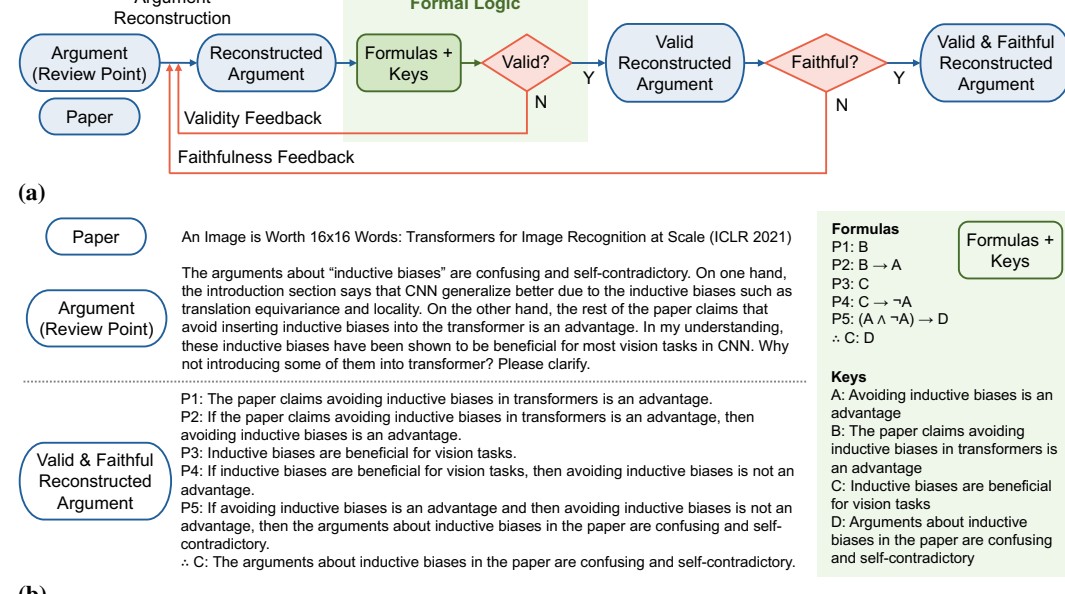

**(a)**

**(b)**

Figure 2: (a) Overview of an automatic argument reconstruction. Given an argumentative review point with a paper, a model first generates a reconstructed argument (i.e., a set of premises and conclusion). To check its *validity*, a model translates a NL reconstructed argument into FOL formulas, and then a SAT solver judges if it is *valid*. To check its *faithfulness*, a model translate FOL formulas back into the NL domain, and a model judges if the reconstruction is *faithful*. If one of two criteria does not met, then corresponding NL feedback is given to the generator model. (b) A representative example. We sample a review point of Dosovitskiy et al. (2020) and its reconstruction along with corresponding formulas and keys.

An LLM alone often falls short in ensuring *validity*, so we include a SAT solver which could automatically judges the *validity* of a set of premises and a conclusion without any logical errors. To do that, an LLM translates a set of natural language (NL) premises and conclusion into first-order logic (FOL) formulas. Then, a SAT solver decides whether the premises deductively implies the conclusion. If the reconstruction is *valid*, then an LLM translates formalized premises and conclusion back into NL domain. This process is called *logical streamlining*, which means to rephrase NL premises or conclusion in order to make their logico-semantic structure more transparent (Bowell & Kemp, 2014; Brun & Betz, 2016; Betz & Richardson, 2021). We then pass these *streamlined* NL premises and conclusion to the subsequent faithfulness feedback loop. However, if the reconstruction is *invalid*, then we feed a rule-based reward signal to the argument reconstructor to regenerate the reconstruction. There are two types of reward signals, one is a naive signal that says the formalized premises do not imply the conclusion, and the other one tells that the proof is circular.

Although the reconstruction is now *valid*, it should also *faithfully* represents the original argument. To ensure that, we prompt an LLM to judge whether the reconstruction is *faithful* or not and justify its decision. If the reconstruction is *faithful*, then we stop iterating the loop. If the reconstruction is *unfaithful*, then we feed LLM judge's justification to the argument reconstructor to regenerate the reconstruction. To minimize model calls in practice, we connect these two loops in series so that only *valid* reconstructions are judged for their *faithfulness*. Also, we limit the total number of loop iteration to 10 and return the last reconstructed argument if the loop fails to generate *valid* and *faithful* reconstruction. We provide an example reconstruction in Figure 2b, and details on feedback loops and model prompts are described in Appendix C and H, respectively.

**Quality of argument reconstruction.** We measure *validity* and *faithfulness* of reconstructed arguments using a SAT solver and human annotators, respectively. With `claude-sonnet-3.7` as a base LLM, every reconstruction is *valid*, and average *faithfulness* score is 4.5 / 5. Detailed analysis including comparison with direct reconstruction is described in Appendix C.

### 3.3 DATASET CONSTRUCTION

Our final destination of proposing REVIEWSCORE is to *automatically* filter out *misinformed* review points using LLMs. To verify LLM's ability to do that, we build a human expert-annotated dataset to measure an agreement between humans and LLMs on REVIEWSCORE evaluation. Our dataset contains total 657 annotated review points, consisting of 143 questions, 92 claims, and 422 arguments. Specifically, 1,748 premises of the arguments are manually annotated [7].

**Human-review matching.** We recruit 15 graduate students studying AI as human annotators, and they annotate total 40 papers submitted to ICLR 2021–2023[8]. Specifically, we first make five groups by their research interests. For each group, three human annotators discuss which papers to annotate and select eight papers that are relevant with all three. Then, each human annotates selected eight papers' review points which are preprocessed from OpenReview.

**Data curation process.** We collaboratively use an LLM and humans for REVIEWSCORE data curation, where an LLM preprocesses reviews and then humans annotate those. Given a review, an LLM extracts independent review points. For each review point, an LLM automatically annotates the type (i.e., claim, argument, or question) and human verifies it. If the review point is a question, then human scores if the question is (un)answerable by the paper in a 5-point scale and justifies it if needed. If the review point is a claim, then human scores if the claim is true in a 5-point scale and justifies it if needed. If the review point is an argument, then human scores the argument's factuality same as in claims (i.e., BASE REVIEWSCORE). To annotate ADVANCED REVIEWSCORE for arguments, we first run automatic argument reconstruction engine (Section 3.2) to extract underlying premises of the argument. After that, human scores if the reconstruction is *faithful* in a 5-point scale. If the *faithfulness* score is less than 4 (i.e., faithful, but one or two minor changes recommended), then they skip the subsequent annotations. Otherwise, they judge the factuality of premises. For each premise, they first select a knowledge base (i.e., submitted paper, annotator knowledge, or referred papers), score if the premise is true based on the knowledge base in a 5-point scale, and score if the premise is (un)trivial in a 3-point scale. They justify any of three decisions if needed.

**Trustworthiness of human annotation.** Three humans independently annotate reviews (i.e., we take a median value as a final human label), and they are allowed to use any related materials or tools (e.g., discussion between authors and reviewers in OpenReview, web search, etc.). We also provide annotators detailed guidelines and an hour-long online orientation session. Furthermore, we highly encourage them to communicate with us through a group chat so that we could give them instant responses to their questions and share with all, which builds a global consensus among human annotators. Thanks to these efforts, despite the difficulty of the work, we obtain median 0.301 and highest 0.428 inter-annotator agreement in Krippendorff's Alpha (Hughes, 2021) across different annotator groups. Detailed results are described in Appendix E.

## 4 RELIABILITY OF AUTOMATIC EVALUATION OF REVIEWSCORE

To evaluate LLM's ability to evaluate REVIEWSCORE, we describe an experimental setup (Section 4.1) and show human-model agreement results (Section 4.2). We also compare the effectiveness of BASE and ADVANCED REVIEWSCORE (Section 4.3). We further analyze human-model disagreements and the effect of providing authors response to models (Section 4.4).

### 4.1 SETUP

Given score rubrics in a 5-point scale, an LLM evaluates REVIEWSCORE according to a knowledge base it selects. We only provide a submitted paper to a model since we assume that the model has a pretrained human-level internal knowledge and has a general understanding of referred papers. Detailed model prompts are described in Appendix H.

---

[7] Details of the dataset construction and statistics are described in Appendix D.
[8] Since ICLR 2024–2025 submission drafts are not opened to public, we exclude these years.

Table 1: Human-model agreement on REVIEWSCORE evaluation.

| Model | ClaimScore | | ArgScore | | QScore | | REVIEWSCORE | |
|---|---|---|---|---|---|---|---|---|
| | F1 | Kappa | F1 | Kappa | F1 | Kappa | F1 | Kappa |
| *Proprietary models* | | | | | | | | |
| claude-sonnet-3.7 | **0.160** | **0.137** | **0.458** | **0.380** | 0.542 | **0.387** | **0.448** | 0.355 |
| claude-sonnet-4 | 0.125 | 0.130 | 0.329 | 0.266 | 0.507 | 0.357 | 0.367 | 0.333 |
| gpt-4o | 0.000 | 0.119 | 0.329 | 0.239 | 0.538 | 0.340 | 0.405 | **0.362** |
| gpt-5 | 0.125 | -0.004 | 0.426 | 0.334 | **0.543** | 0.300 | 0.441 | 0.310 |
| gemini-2.5-flash | 0.091 | 0.106 | 0.447 | 0.366 | 0.512 | 0.225 | 0.439 | 0.318 |
| *Open-sourced models* | | | | | | | | |
| qwen3-235b-a22b | **0.231** | 0.094 | **0.403** | 0.149 | 0.531 | 0.234 | **0.436** | 0.243 |
| llama-3.3 | 0.133 | 0.153 | 0.250 | 0.107 | 0.551 | **0.284** | 0.409 | **0.355** |
| deepseek-v3 | 0.000 | **0.165** | 0.276 | **0.196** | **0.597** | 0.228 | 0.382 | 0.310 |

**Language models.** To measure LLM's ability to automatically evaluate REVIEWSCORE, we perform experiments on eight current state-of-the-art LLMs that achieve significantly high alignments with humans. We include five proprietary models, claude-sonnet-3.7 (Anthropic, 2025a), claude-sonnet-4 (Anthropic, 2025b), gpt-4o (OpenAI, 2024), gpt-5 (OpenAI, 2025), and gemini-2.5-flash (Gemini Team, Google, 2025), and three open-sourced models, qwen3-235b-a22b (Yang et al., 2025), deepseek-v3 (DeepSeek-AI et al., 2024), and llama-3.3 (AI@Meta, 2024)[9]. To get consistent and reliable scores from LLM judges, we set a low temperature (i.e., 0) and select the highest probability response from the model (Liang et al., 2022; Liu et al., 2023; Gu et al., 2024). We exclude reasoning models as LLM judges since REVIEWSCORE mostly depends on grounding and evidence, not longer or smarter chains of thought.

**Evaluation metrics.** We use different sets of metrics for two types of problem formulations, binary classification and 5-point scale scoring. For the binary classification, since majority of human-annotated labels are *Not misinformed*, we mainly use F1 score which is robust to class imbalance. For the 5-point scale scoring, since majority of human-annotated scores are 4 and 5, we mainly use Quadratic Weighted Kappa (Warrens, 2012), a variant of Cohen's Kappa (Cohen, 1960), that is robust to the skewed data distribution. Hereinafter, we call this metric Kappa for simplicity. To provide more comprehensive results, we additionally use Precision and Recall for the binary classification and Pearson rank correlation and Gwet's AC2 (Gwet, 2001) for the 5-point scale scoring.

## 4.2 MAIN RESULTS

We empirically validate the alignment of human-annotated and model-estimated REVIEWSCORE using different models and evaluation metrics in Table 1. Most models show 0.4–0.5 F1 score and 0.3–0.4 Kappa score, which indicates moderate agreement between humans and models on REVIEWSCORE evaluation. However, there are differences in human-model agreements for three types of review points. Regardless of the models, questions show the highest agreement, arguments follow subsequently, and claims show the lowest agreement. Specifically, for claims, some models show zero F1 score or negative Kappa score. We analyze human-model disagreements thoroughly in Section 4.4 and conclude that since claims lack supporting evidence and are often value-laden, models often misinterpret the intended meaning of the claims or judge differently than humans.

For proprietary models, claude-sonnet-3.7 shows the highest human-model agreements across different types of review points. One thing to note is claude-sonnet-4 consistently shows lower agreement than its preceding model. We find out that it's because claude-sonnet-4 sometimes overthink and more frequently predicts the minority human-annotated scores than claude-sonnet-3.7 does. In contrast, comparing F1 scores of gpt-4o and gpt-5, the advanced model shows higher agreement than the preceding model.

For open-sourced models, qwen3-235b-a22b shows best human-model agreements in terms of F1 score. Open-sourced models show lower human-model agreements than proprietary models, but the gap is not significant. Open-sourced models sometimes show better agreements than proprietary models, for instance, llama-3.3 outperforms claude-sonnet-4 on REVIEWSCORE evalua-

---

[9]Details of model specifications are described in Appendix G

Table 2: Comparison of human-model agreement of BASE vs. ADVANCED REVIEWSCORE.

| Model | Metric | Base | Advanced w/o Agg | Advanced |
|---|---|---|---|---|
| claude-sonnet-3.7 | F1 | 0.185 | 0.262 | **0.458** |
| | Kappa | 0.153 | 0.280 | **0.380** |
| gpt-5 | F1 | 0.243 | 0.316 | **0.426** |
| | Kappa | 0.223 | 0.315 | **0.334** |
| gemini-2.5-flash | F1 | 0.167 | 0.349 | **0.447** |
| | Kappa | 0.138 | 0.330 | **0.366** |
| deepseek-v3 | F1 | 0.203 | 0.163 | **0.276** |
| | Kappa | 0.120 | **0.202** | 0.196 |

**(a)**                                    **(b)**

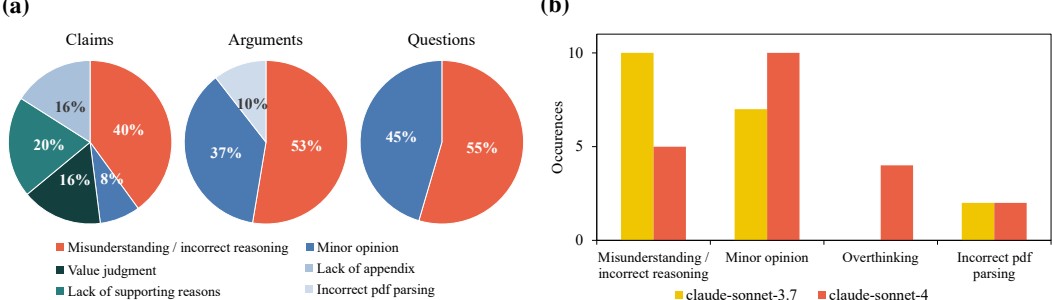

Figure 3: Human-model disagreement analysis. (a) Types of human-model disagreements. (b) Comparison of disagreement types of claude-sonnet-3.7 and claude-sonnet-4.

tion. However, open-sourced models consistently show lower agreements on evaluating ArgScore. More results and qualitative analysis of model evaluation are described in Appendix F.

## 4.3 BASE VS. ADVANCED REVIEWSCORE

We compare human-model agreements of BASE and ADVANCED REVIEWSCORE using different models and evaluation metrics. Since the only difference is in defining scores for arguments, we compare REVIEWSCORE for arguments in Table 2. To verify the effectiveness of aggregation methods, we additionally include ADVANCED REVIEWSCORE for arguments without aggregation (i.e., factuality of premises). Regardless of models, ADVANCED REVIEWSCORE clearly shows higher agreements than BASE REVIEWSCORE. For claude-sonnet-3.7, ADVANCED REVIEWSCORE performs 2.48× higher F1 score and 2.48× higher Kappa score than the BASE REVIEWSCORE. The aggregation methods also contribute to higher agreements. For claude-sonnet-3.7, ADVANCED REVIEWSCORE performs 1.75× higher F1 score and 1.36× higher Kappa score than the one without aggregation.

## 4.4 ANALYSIS

**REVIEWSCORE Distribution.** We present model-estimated REVIEWSCORE distribution and compare with human-annotated ones. Specifically, following the binary classification setup, we report the proportion of *misinformed* review points in Table 3. There is a clear trend that models more frequently decide review points as *Misinformed* than humans. Models decide 31.2% of reviews points as *Misinformed* in average, where humans decide only 17.8% as *Misinformed*. The discrepancy is the largest in claims where models decide 22.8% as *Misinformed* and humans decide only 5.9% as *Misinformed*. To understand these results more deeply, we continue our analysis on the following human-model disagreements.

Table 3: Proportion of *misinformed* review points estimated by models and humans.

| | Model | Human |
|---|---|---|
| Claims | 22.8 | 5.9 |
| Arguments | 30.1 | 17.7 |
| Questions | 40.0 | 26.4 |
| Review points | 31.2 | 17.8 |

**Human-model disagreements.** We analyze types of human-model disagreements in Figure 3a. Across all review points, there are two common types of disagreements, which are models misunderstand or incorrectly reason and models predict scores that are minority of human-annotated

Table 4: Effect of providing Authors Response (AR) to a model for REVIEWSCORE evaluation.

|  | F1 | | Kappa | |
|---|---|---|---|---|
|  | w/o AR | w/ AR | w/o AR | w/ AR |
| WScore | 0.211 | **0.286** | 0.178 | **0.212** |
| ArgScore w/o Agg | 0.230 | **0.283** | 0.271 | **0.312** |
| ArgScore | 0.329 | **0.447** | 0.266 | **0.340** |
| QScore | 0.507 | **0.513** | **0.357** | 0.324 |
| ClaimScore | 0.125 | **0.133** | 0.130 | **0.225** |
| REVIEWSCORE | 0.367 | **0.449** | 0.333 | **0.377** |

scores. These two types comprise nearly all disagreements on arguments and questions, and about half of disagreements on claims. For claims, we observe that 36% of disagreements are caused by either claims are value-laden or lack supporting reason. These make the factuality judgment of claims subjective, leading to a low human-model agreement. Practical limitations such as incorrect pdf parsing and not providing appendix to models cause nontrivial portion of disagreements. We also analyze the reason why `claude-sonnet-3.7` shows higher human-model agreements than `claude-sonnet-4` in Figure 3b. The advanced model shows smaller number of misunderstanding or incorrect reasoning, however, it shows higher number of overthinking[10] and minority human-annotated score predictions, leading to a lower human-model agreement.

**Helpfulness of authors response.** We study whether providing authors responses of reviews to a model benefits the automatic REVIEWSCORE evaluation using `claude-sonnet-4` in Table 4. Providing authors response clearly shows higher human-model agreement than not providing it. Specifically, the agreement of REVIEWSCORE increases by 22.3% in F1 score and 13.2% in Kappa score by incorporating authors response. Note that `claude-sonnet-4` with authors response achieves the highest agreement compared to all models without authors response. Furthermore, the agreement of WScore and ArgScore without aggregation (i.e., premise factuality) also increases significantly by incorporating authors response. These observations show that authors response largely benefits detecting *misinformed* reviews by providing additional cues for model judgments.

## 5 LIMITATIONS

We acknowledge three types of limitations of our work. First, there are technical and practical limitations in the automated REVIEWSCORE evaluation. Throughout Section 4, human-model agreement seems low even with authors response. However, compared to Du et al. (2024) which measures human-model agreement on *Deficient* review detection and reports the best F1 score as 0.220, our work reports the best F1 score as 0.449, indicating more than $2\times$ agreement. To save API calling costs, when the paper is given to LLMs, we only provide text and tables of the main paper, where any figure or appendix is excluded. We also observe nontrivial amount of incorrect pdf parsing. Second, there are technical limitations in the automatic argument reconstruction. The reconstruction output is not always perfect since it depends on the capability of the base model. However, we observe a considerable output quality improvement when we upgrade the base model to `claude-sonnet-4`, indicating that the reconstruction engine would perform better as the model improves in general. Lastly, there are practical limitations in the dataset construction. Since human annotators are graduate students with varying skillfulness and the annotation requires significant amount of cognitive load, there are unavoidable noise in human annotation. Furthermore, a manual selection of papers by human annotators might introduce any kind of unintended biases. However, to collect the most reliable annotation under our limited budget, we inevitably choose this method.

## 6 CONCLUSION

We introduce REVIEWSCORE, a new evaluation of peer review quality, focusing on detecting *misinformed* review points. To automate REVIEWSCORE evaluation using LLMs and validate it, we also

---

[10] We judge a model overthinks if its justification of a score is significantly longer than another model. We also manually verify if a model overanalyzes a review point.

construct a trustworthy human-annotated dataset. The results show a moderate human-model agreement, and further comprehensive disagreement analysis supports the potential of a fully automated REVIEWSCORE evaluation with current state-of-the-art LLMs. We hope our work could contribute to mitigate the issue of low-quality reviews in AI conferences.

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

## A    FUTURE IMPACT

We expect automatic REVIEWSCORE evaluation could greatly benefit different roles of the current peer reviewing system. For authors, by providing reconstruction of argumentative review points, it helps to understand or clarify the reviewer's intention and to formulate a rebuttal. For reviewers, by providing REVIEWSCORE of their review points, it allows them to verify the review quality by themselves and helps reviewers to better understand the paper. For metareviewers, by providing REVIEWSCORE of each reviewer, it could assist their final decisions. To summarize, it could serve as an automated system for managing the review quality.

## B    REVIEW QUALITY CRITERIA DISCUSSION

First, we recruit a group of three graduate students studying AI and NLP and let them independently analyze reviews of ten common manually selected submitted papers from ICLR 2021–2023[11]. Specifically, we guided the group to: decompose a review into several independent review points, evaluate quality of the review points by a 5-point scale with their own criteria, and justify their scores that ends with a meta-sentence that does not involve paper's context which is used for our further analysis. During human evaluation, we minimize our effort to provide detailed guidelines to facilitate bias-free human analysis.

Then, we categorize these justifications in order to find common features of low-quality review points. There are five common types: (1) questions that can be already addressed by the paper, (2) comments that reflect a misunderstanding of the paper, (3) out-of-scope remarks, (4) observations pointing out minor details, and (5) unclear points. However, for the last three types, we observe that majority of review points are agreed or argued by a single human annotator. In other words, given a review point, only a single human argues it is out-of-scope, whereas other two humans argue it is within scope and could be a potential drawback of the submitted paper. Following this pattern, a single human argues a review point is addressing minor details but others argue it is a major point, and a single human argues a review point is unclear whereas others do not agree with this. In contrast, the first two types of review points (i.e., answerable questions and misunderstood comments) are mostly agreed by two or more human annotators, indicating that these two types have more objective and trustworthy criteria for detecting low-quality reviews. Based on these observations, we focus on evaluating review points based on the first two criteria.

Lastly, we share some meaningful insights from this group discussion that strongly motivates our work. Based on the analysis of score justifications, given a review point, some (sub)sentences are high-quality or factually correct but others could be misinformed or factually incorrect. This later motivates our premise-level factuality evaluation (Section 3.1). Furthermore, during the group discussion, we observe that human annotators sometimes struggle which parts in a review point they should weigh more to evaluate the review point. This later motivates our aggregation methods which include logical conjunction and weighted average by untrivialness (Section 3.1). We also discover that human evaluations are sometimes incorrect. We leave this as a limitation of our work. However, to alleviate this issue, we recruit three human annotators for every instance and further ensure the annotation quality by providing them careful guidelines and actively communicating with them to build a global consensus on the evaluation criteria (Section 3.3).

---

[11]Since those papers include papers written by authors of our work, we do not share the full list of target papers to keep anonymity. We will uncover the full list after the paper is published.

## C  AUTOMATIC ARGUMENT RECONSTRUCTION DETAILS AND RESULTS

### C.1  IMPLEMENTATION DETAILS

We elaborate components of automatic argument reconstruction engine (Section 3.2, Figure 2) in the following. We also refer the corresponding prompt used in each step if it exists.

1. Given an argumentative review point, an LLM extracts a verbatim conjecture and its verbatim reason statements (Figure 23).

2. Given a verbatim conclusion and reasons of the argumentative review point and a corresponding paper parsed from a pdf file, an LLM reconstructs the argument into a premise-conclusion structure. At the same inference, the model also translates (or formalizes) NL premises and conclusion into corresponding FOL formulas and generate keys which assign NL meaning to variables and predicates. To facilitate the model to generate a *valid* formalization, the model generates a deductive proof using formalized premises and conclusion at the end (Figure 24).

3. Given FOL premises and conclusion with keys and a deductive proof, an LLM extracts necessary FOL premises for the deductive proof, write a python program using Z3 (De Moura & Bjørner, 2008) that automatically checks the *validity* of the necessary FOL premises and conclusion, extracts a final FOL conclusion that is used in the python program, and judges whether the proof is circular (i.e., whether the final FOL conclusion is included in one of necessary FOL premises.) (Figure 25, Figure 26).

4. If the proof is circular, then a NL feedback indicating circularity of the proof is sent to Step 2, and the model re-generates an argument reconstruction. Otherwise, we run the python program that checks the *validity* of the reconstruction. However, if the program returns an error, the model takes this error message and re-generate the python program that fixes the error (Figure 27). If the reconstruction is *invalid*, then a NL feedback indicating *invalidity* of the reconstruction is sent to Step 2, and the model re-generates an argument reconstruction. Otherwise (i.e., if the reconstruction is *valid*), we proceed to the next step.

5. To check *faithfulness* of the reconstruction, the model first translates FOL premises and conclusion (one of the outputs of the model in Step 2) with keys (one of the outputs of the model in Step 1) into NL premises and conclusion (Figure 28). This process is called *logical streamlining* in logic and critical thinking (Bowell & Kemp, 2014; Brun & Betz, 2016; Betz & Richardson, 2021).

6. Lastly, given an original argumentative review point (or an argument) and the *streamlined* NL premises and conclusion, the model judges whether the reconstruction is *faithful* with justifications (Figure 29). If the reconstruction is *unfaithful*, then a NL feedback including the model's justifications is sent to Step 2, and the model re-generates an argument reconstruction accordingly. Otherwise, since the reconstruction is *valid* yet *faithful*, the feedback loop is finished and the *streamlined* NL premises and conclusion become a final argument reconstruction.

## C.2 QUANTITATIVE RESULTS

To verify effectiveness of the feedback loop, we report the argument reconstruction performance with and without feedback in Table 5. We mainly measure average *validity* and *faithfulness* of the reconstructed arguments using a SAT solver and human annotators, respectively. We provide score rubric for evaluating *faithfulness* in Figure 4. We also include the pass rate, which indicates whether the (last) reconstructed argument fulfill the *validity* and *faithfulness* criteria in the feedback loop. Furthermore, we report the average number of loop iterations to check if the feedback loops are actively used.

In Table 5, we verify that the proposed method (i.e., w/ feedback) achieves a perfect *validity* and nearly perfect *faithfulness* and pass rate, whereas the performance of the direct reconstruction (i.e., w/o feedback) largely lags behind that. We observe that the average number of loop iterations is 3.09, indicating active usage of feedback signals, but the number varies a lot depending on the clarity of logical structures of the arguments.

Table 5: Performance of Automatic Argument Reconstruction.

|              | w/o feedback | w/ feedback      |
| ------------ | ------------ | ---------------- |
| Validity     | 0.895        | **1.00**         |
| Faithfulness | 2.91         | **4.47**         |
| Pass Rate    | 0.369        | **0.935**        |
| # of loops   | N/A          | $3.09 \pm 2.73$  |

**Faithfulness Score Rubric for Argument Reconstruction**

5: Definitely faithful, no change is essential
4: Faithful, but one or two minor details in premises need to change in order to fully expres s the original context
3: Faithful, but more than two minor details in premises need to change
2: Not faithful, one or two major error exist
1: Definitely not faithful, more than two major errors exist

Figure 4: Score rubric for evaluating *faithfulness* of argument reconstruction by human annotators.

## C.3 QUALITATIVE RESULTS

We present three representative examples of model-generated argument reconstruction and compare outputs with and without the feedback loop. We observe the direct reconstruction (i.e., w/o feedback) often generates either *invalid* or *unfaithful* reconstructed arguments, however, the proposed method (i.e., w/ feedback) generates *valid* yet *faithful* reconstructed arguments. We provide detailed analysis in Figure 5–7.

**Input: Argument (Review Point) & Paper**

*Paper*:
Least-to-Most Prompting Enables Complex Reasoning in Large Language Models (Zhou et al., 2023a)

------------------------------------------------------------------------------------------------------

*Argument (Review Point)*:
The paper lacks insights on key questions. Does the model have ability to decompose questions very different from demo examples into correct sub-problems? Even the model solves each sub-problem correctly, what makes the model compose the final solution correctly?

**Output: Reconstructed Argument**

*Direct Reconstruction (w/o feedback)*:
P1: The paper does not demonstrate whether the model has the ability to decompose questions very different from demo examples into correct sub-problems.
P2: The paper does not explain what makes the model compose the final solution correctly even when it solves each sub-problem correctly.
P3: If a paper fails to provide insights on key questions about a proposed method, then the paper lacks insights on key questions.
P4: Questions about decomposition ability and solution composition are key questions for the least-to-most prompting method.
∴ C: The paper lacks insights on key questions.

------------------------------------------------------------------------------------------------------

*Proposed Method (w/ feedback)*:
P1: The paper does not demonstrate whether the model has the ability to decompose questions very different from demo examples into correct sub-problems.
P2: If the paper does not demonstrate whether the model has the ability to decompose questions very different from demo examples into correct sub-problems, then the paper fails to provide insights on the key question about decomposition ability.
P3: If the paper fails to provide insights on the key question about decomposition ability or fails to provide insights on the key question about solution composition, then the paper fails to provide insights on key questions.
P4: If the paper fails to provide insights on key questions, then the paper lacks insights on key questions.
∴ C: The paper lacks insights on key questions.

**Analysis**

 - For the direct reconstruction, P1, P2, P4 semantically implies an intermediate conclusion that "The paper fails to provide insights on key questions about the proposed method.", but this connecting premise is not explicitly reconstructed. Therefore, this reconstruction is *invalid* [✗]. However, if we consider that this connecting premise is implicitly presumed, then the reconstruction *faithfully* [✓] represents the original argument.
 - For the proposed method, the reconstruction is logically *valid* [✓] yet *faithfully* [✓] represents the original argument.

Figure 5: Example #1 of automatic argument reconstruction.

---

**Input: Argument (Review Point) & Paper**

*Paper*:
ReAct: Synergizing Reasoning and Acting in Language Models (Yao et al., 2023)

---

*Argument (Review Point)*:
ReAct alone works fairly well for some tasks, but self-consistent Chain of Thought often outperforms ReAct alone, and leads to fairly dramatic increases in performance. The best models use a smart combination of both CoT and ReAct, using heuristics that likely maximize performance on the dataset(s) (see 'ReAct → CoT–SC' strategy), which makes the performance of ReAct by itself slightly less impressive (though I still think this is a useful contribution for the community)

---

**Output: Reconstructed Argument**

*Direct Reconstruction (w/o feedback)*:
P1: The best models use a smart combination of both CoT and ReAct.
P2: Models use heuristics that likely maximize performance on the dataset(s) (see 'ReAct → CoT–SC' strategy).
P3: If models use heuristics that likely maximize performance on the dataset(s), then the performance of ReAct by itself is less impressive.
P4: If the best models use a combination of both CoT and ReAct rather than ReAct alone, then the performance of ReAct by itself is less impressive.
P5: The performance of ReAct by itself being less impressive still means it is a useful contribution for the community.
∴ C: ReAct alone works fairly well for some tasks, but self-consistent Chain of Thought often outperforms ReAct alone, and leads to fairly dramatic increases in performance.

---

*Proposed Method (w/ feedback)*:
P1: ReAct alone works fairly well for some tasks.
P2: Self-consistent Chain of Thought often outperforms ReAct alone.
P3: The best models use a smart combination of both CoT and ReAct.
P4: Models use heuristics like the 'ReAct → CoT–SC' strategy that maximize performance.
P5: If Self-consistent Chain of Thought often outperforms ReAct alone, and the best models use a smart combination of both CoT and ReAct, and models use heuristics like the 'ReAct → CoT–SC' strategy that maximize performance, then the performance of ReAct by itself is slightly less impressive.
P6: If ReAct alone works fairly well for some tasks, and the performance of ReAct by itself is slightly less impressive, and a technique can be a useful contribution even if its performance is less impressive, then ReAct is still a useful contribution for the community.
P7: A technique can be a useful contribution even if its performance is less impressive.
∴ C: The performance of ReAct by itself is slightly less impressive, and ReAct is still a useful contribution for the community.

---

**Analysis**

- For the direct reconstruction, both pairs of premises, P1 & P4, and P2 & P3, deductively imply an intermediate conclusion that "The performance of ReAct by itself is less impressive.". However, there is no logical connection between this intermediate conclusion and the final conclusion C, which indicates the reconstruction is *invalid* [✗]. Furthermore, the final conclusion C does not correctly represent the original argument, which means the reconstruction is *unfaithful* [✗].
- For the proposed method, the premises deductively imply the final conclusion (i.e., *valid* [✓]), and the reconstruction correctly yet completely represents the original argument (i.e., *faithful* [✓]), including the correct final conclusion.

Figure 6: Example #2 of automatic argument reconstruction.

**Input: Argument (Review Point) & Paper**

*Paper*:
Automatic Chain of Thought Prompting in Large Language Models (Zhang et al., 2023)

---

*Argument (Review Point)*:
Finally, if clustering was the way to go, the authors could have tried to cluster samples into $n$-clusters, see where the test sample is fitting, and could have tried to use just one prompt instead of $k$ different ones at each time. This might have reduced the compute cost and would be interesting to see the performance vs resource trade-off.

**Output: Reconstructed Argument**

*Direct Reconstruction (w/o feedback)*:
P1: The authors could have tried to cluster samples into n-clusters and see where the test sample is fitting.
P2: The authors could have tried to use just one prompt instead of $k$ different ones at each time.
P3: Using one prompt instead of $k$ different ones would reduce the compute cost.
P4: If a method reduces compute cost, it would be interesting to see the performance vs resource trade-off.
P5: If the authors could have done something that would be interesting to see, then they should have done it.
∴ C: The authors should have tried to cluster samples into $n$-clusters, see where the test sample is fitting, and use just one prompt instead of $k$ different ones at each time.

---

*Proposed Method (w/ feedback)*:
P1: An alternative approach would be to cluster samples into $n$-clusters, see where the test sample is fitting, and use just one prompt instead of $k$ different ones.
P2: If an alternative approach would use just one prompt instead of $k$ different ones, then the alternative approach would reduce compute cost.
P3: If the alternative approach would reduce compute cost, then examining the performance vs. resource trade-off of the alternative approach would be interesting.
P4: If an alternative approach would use just one prompt instead of $k$ different ones and examining the performance vs. resource trade-off of the alternative approach would be interesting, then the alternative approach would have been interesting to evaluate for its performance vs. resource trade-off.
∴ C: The alternative approach would have been interesting to evaluate for its performance vs. resource trade-off.

**Analysis**

 - For the direct reconstruction, P3 and P4 imply an intermediate conclusion that "Using one prompt instead of k different ones would be interesting to see the performance vs resource trade-off.". Let us call this IC1. IC1 and P5 imply another intermediate conclusion that "The authors should have tried using one prompt instead of k different ones.". Let us call this IC2. Lastly, P1, P2, and IC2 imply the final conclusion C, indicating the reconstruction is *valid* [✓]. However, the final conclusion is stronger than what the original argument states. The original argument suggests an alternative approach, but the reconstructed final conclusion obligates authors to try it, meaning the reconstruction is *unfaithful* [✗].
 - For the proposed method, the reconstruction is *valid* [✓] and *faithful* [✓], including the correct degree of strength of the final conclusion.

Figure 7: Example #3 of automatic argument reconstruction.

# D  DATASET DETAILS

## D.1  TERMS OF USE AND LICENSE

All papers and corresponding reviews used in our work are crawled from OpenReview[12], and our work is consistent with the Openreview terms of use[13]. Following these terms, we will release our work with a CC-BY 4.0 license.

## D.2  PILOT DATASET

Before we construct the main dataset in Section 3.3, to ensure a trustworthiness and reliability of human annotation, we conduct a pilot study. We recruit three graduate students studying AI as human annotators, and let them choose total five papers submitted to ICLR 2021–2025[14]. Specifically, each human annotates reviews of three papers in OpenReview, where only one paper is authored by themselves and the other two non-authored papers are assigned in common. To ensure trustworthiness of non-authors' annotation, three humans annotate reviews and then we take a median value as a final human label.

## D.3  DATASET STATISTICS

We present the REVIEWSCORE dataset statistics including the pilot and main subsets in Table 6. We include number of instances and percentage of *misinformed* labels.

Table 6: REVIEWSCORE dataset statistics.

|  | Pilot | Main | Total |
|---|---|---|---|
| ICLR Years | 2021–2025 | 2021–2023 | - |
| # Papers | 5 | 40 | 45 |
| # Review(er)s | 19 | 155 | 174 |
| *Number of instances* | | | |
| Review Points | 84 | 573 | 657 |
| Questions | 22 | 121 | 143 |
| Claims | 5 | 87 | 92 |
| Arguments | 57 | 365 | 422 |
| Premises | 227 | 1,521 | 1,748 |

## D.4  FULL LIST OF PAPERS

We provide a full list of papers used in the main subset of REVIEWSCORE dataset in Table 7. As explained in Section 3.3, eight papers are selected by each human annotator group with a common research interest.

# E  ANALYSIS OF HUMAN ANNOTATORS

## E.1  HUMAN ANNOTATOR EXPERTISE

We report human annotator's expertise on REVIEWSCORE evaluation in Table 8. Specifically, for each annotator, we indicate a number of publications in AI/ML (including arXived works) and an averaged paper relevance score across eight assigned papers. We also indicate averaged numbers and scores for each group. The results show that an average number of publication is 3.93 and average paper relevance score is 4.06 / 5, indicating highly-experienced and relevant experts conduct a human annotation process. However, we observe that there are inter-group gaps in human expertise. Specifically, Group 2 and 4 shows significantly higher number of publications and paper relevance

---

[12]https://openreview.net
[13]https://openreview.net/legal/terms
[14]Unlike the main dataset, we include ICLR 2024–2025 since authors can access to their own submissions.

Table 7: Full list of ICLR submitted papers used in the REVIEWSCORE dataset.

| Group | Topic | Paper Title |
|---|---|---|
| 1 | Image Generation | Lipman et al. (2023), Hertz et al. (2023), Couairon et al. (2023), Singer et al. (2023), Meng et al. (2022), Gal et al. (2023), Poole et al. (2023), Kim et al. (2023) |
| 2 | Time Series ML | Shukla & Marlin (2021), Zhang et al. (2022), Borisov et al. (2023), Nie et al. (2023), Nam et al. (2023), Zhang & Yan (2023), Wang et al. (2023), Shukla & Marlin (2022) |
| 3 | LLM Reasoning / Compression | Hsu et al. (2022), Ji et al. (2023), Lu et al. (2023), Ni et al. (2023), Fu et al. (2023), Liang et al. (2021), Ruis et al. (2023), Arora et al. (2023) |
| 4 | LLM / VLM | Hu et al. (2022), Kukleva et al. (2023), Wei et al. (2022), Sanh et al. (2022), Shen et al. (2022), Zhou et al. (2023b), Si et al. (2023), Shi et al. (2023) |
| 5 | LLM Prompting | Zhang et al. (2023), Yao et al. (2023), Zhou et al. (2023a), Razdaibiedina et al. (2023), Li et al. (2022), Jang et al. (2022), Creswell et al. (2023), Huang et al. (2023) |

than other groups. Detailed score rubric for paper relevance is described in Figure 8, and we note that there is no instance where paper relevance of any human annotator is less than 3 (i.e., Moderate relevance).

Table 8: Human annotator's expertise on REVIEWSCORE evaluation. A **bold** indicates the highest number/score across different groups, and an underline indicates the second highest.

| Group | Annotator ID | # AI/ML Pub | Paper Relevance | Avg # Pub | Avg Paper Relevance |
|---|---|---|---|---|---|
| 1 | anno_11 | 2 | 3.50 | 3.00 | 4.00 |
| | anno_12 | 3 | 4.13 | | |
| | anno_13 | 4 | 4.38 | | |
| 2 | anno_21 | 6 | 4.50 | 5.33 | **4.38** |
| | anno_22 | 3 | 4.38 | | |
| | anno_23 | 7 | 4.25 | | |
| 3 | anno_31 | 0 | 4.00 | 1.33 | 3.96 |
| | anno_32 | 1 | 4.00 | | |
| | anno_33 | 3 | 3.88 | | |
| 4 | anno_41 | 1 | 4.00 | **6.67** | 4.04 |
| | anno_42 | 6 | 3.88 | | |
| | anno_43 | 13 | 4.25 | | |
| 5 | anno_51 | 3 | 3.13 | 3.33 | 3.92 |
| | anno_52 | 1 | 4.13 | | |
| | anno_53 | 6 | 4.50 | | |
| Total | - | - | - | 3.93 | 4.06 |

---

**Paper Relevance Score Rubric**

5: Direct expertise – Works in the exact subtopic and can judge nuanced claims, methods, and datasets.
4: Strong relevance – Adjacent/overlapping subtopic with regular use of the paper's methods or domain; can evaluate technical choices with minimal ramp-up.
3: Moderate relevance – Same broad area (e.g., NLP $\leftrightarrow$ NLP; CV $\leftrightarrow$ CV) but different subtopic or methods; will understand contributions but may miss edge-case nuances.
2: Low relevance – Only tangential connection (e.g., general ML experience while the paper is domain-specific) and limited familiarity with core methods or domain.
1: No clear relevance – Outside the field; would require substantial background reading to assess claims/methodology.

---

Figure 8: Score rubric for evaluating paper relevance of human annotators.

### E.2 INTER-ANNOTATOR AGREEMENT

To ensure trustworthiness of human annotation, we report inter-annotator agreement in Krippendorff's Alpha (Hughes, 2021) on REVIEWSCORE evaluation in Table 9. Overall, REVIEWSCORE shows 0.301 Krippendorff's Alpha, indicating a moderate inter-annotator agreement. Specifically, ClaimScore shows the highest agreement, QScore follows subsequently, and ArgScore shows the lowest agreement.

By comparing inter-group agreements, Group 2 and 4 show significantly higher REVIEWSCORE agreement than other groups. In contrast, Group 1 and 5 include negative inter-annotator agreement, which is a critical concern for reliable human annotation. By a manual disagreement analysis, we confirm that most disagreements come from human annotators with low paper relevance (i.e., anno_11 and anno_51 in Table 8). To summarize, we observe that the inter-annotator agreement aligns with the annotator expertise (Table 8), and we (unsurprisingly) confirm that human annotations could be more reliable if their research interests become more relevant to the assigned papers.

Table 9: Inter-annotator agreement (Krippendorff's Alpha) on REVIEWSCORE evaluation. A **bold** indicates the highest agreement across different groups, and an underline indicates the second highest.

| Group | ClaimScore | ArgScore | QScore | REVIEWSCORE |
|---|---|---|---|---|
| 1 | **0.399** | -0.023 | 0.245 | 0.125 |
| 2 | 0.328 | 0.329 | 0.350 | **0.428** |
| 3 | **0.400** | 0.146 | 0.436 | 0.301 |
| 4 | 0.375 | **0.361** | **0.465** | 0.385 |
| 5 | 0.203 | -0.023 | -0.078 | -0.041 |
| Median | 0.375 | 0.146 | 0.350 | 0.301 |

## F ADDITIONAL RESULTS OF AUTOMATIC REVIEWSCORE EVALUATION

### F.1 QUANTITATIVE RESULTS

We additionally report human-model agreement on REVIEWSCORE evaluation using different evaluation metrics in Table 10–13. Specifically, we report agreement on ClaimScore evaluation in Table 10, agreement on ArgScore evaluation in Table 11, agreement on QScore evaluation in Table 12, and agreement on REVIEWSCORE evaluation in Table 13. For the binary classification setup, we use Precision, Recall, and F1 Score, and for the 5-point scale setup, we use Pearson rank correlation, Gwet's AC2 (Gwet, 2001), and Quadratic Weighted Kappa (Warrens, 2012).

Table 10: Human-model agreement on ClaimScore evaluation.

| Model | Binary | | | 5-point Scale | | |
|---|---|---|---|---|---|---|
| | Precision | Recall | F1 | Pearson | AC2 | Kappa |
| *Proprietary models* | | | | | | |
| claude-sonnet-3.7 | **0.091** | **0.667** | **0.160** | **0.192** | 0.035 | **0.137** |
| claude-sonnet-4 | **0.091** | 0.200 | 0.125 | 0.172 | 0.048 | 0.130 |
| gpt-4o | 0.000 | 0.000 | 0.000 | 0.130 | **0.064** | 0.119 |
| gpt-5 | 0.083 | 0.250 | 0.125 | -0.014 | -0.057 | -0.004 |
| gemini-2.5-flash | 0.059 | 0.200 | 0.091 | 0.146 | 0.054 | 0.106 |
| *Open-sourced models* | | | | | | |
| qwen3-235b-a22b | **0.136** | **0.750** | **0.231** | 0.135 | 0.045 | 0.094 |
| llama-3.3 | 0.091 | 0.250 | 0.133 | 0.144 | **0.122** | 0.153 |
| deepseek-v3 | 0.000 | 0.000 | 0.000 | **0.195** | 0.086 | **0.165** |

Table 11: Human-model agreement on ArgScore evaluation.

| Model | Binary | | | 5-point Scale | | |
|---|---|---|---|---|---|---|
| | Precision | Recall | F1 | Pearson | AC2 | Kappa |
| *Proprietary models* | | | | | | |
| claude-sonnet-3.7 | 0.345 | **0.679** | **0.458** | **0.425** | **0.369** | **0.380** |
| claude-sonnet-4 | 0.250 | 0.482 | 0.329 | 0.258 | 0.239 | 0.266 |
| gpt-4o | **0.483** | 0.250 | 0.329 | 0.280 | 0.221 | 0.239 |
| gpt-5 | 0.353 | 0.536 | 0.426 | 0.368 | 0.329 | 0.334 |
| gemini-2.5-flash | 0.333 | **0.679** | 0.447 | 0.419 | 0.358 | 0.366 |
| *Open-sourced models* | | | | | | |
| qwen3-235b-a22b | **0.381** | 0.429 | 0.403 | 0.281 | 0.012 | 0.149 |
| llama-3.3 | 0.344 | 0.196 | 0.250 | -0.014 | 0.124 | 0.107 |
| deepseek-v3 | 0.267 | 0.286 | 0.276 | 0.238 | **0.188** | **0.196** |

Table 12: Human-model agreement on QScore evaluation.

| Model | Binary | | | 5-point Scale | | |
|---|---|---|---|---|---|---|
| | Precision | Recall | F1 | Pearson | AC2 | Kappa |
| *Proprietary models* | | | | | | |
| claude-sonnet-3.7 | **0.571** | 0.516 | **0.542** | **0.394** | **0.380** | **0.387** |
| claude-sonnet-4 | 0.462 | 0.563 | 0.507 | 0.387 | 0.335 | 0.357 |
| gpt-4o | 0.447 | 0.677 | 0.538 | **0.395** | 0.298 | 0.340 |
| gpt-5 | 0.500 | 0.594 | **0.543** | 0.320 | 0.285 | 0.300 |
| gemini-2.5-flash | 0.386 | **0.759** | 0.512 | 0.283 | 0.154 | 0.225 |
| *Open-sourced models* | | | | | | |
| qwen3-235b-a22b | 0.394 | 0.813 | 0.531 | 0.325 | 0.117 | 0.234 |
| llama-3.3 | 0.409 | **0.844** | 0.551 | **0.375** | **0.196** | **0.284** |
| deepseek-v3 | **0.469** | 0.821 | **0.597** | 0.325 | 0.101 | 0.228 |

Table 13: Human-model agreement on REVIEWSCORE evaluation.

| Model | Binary | | | 5-point Scale | | |
|---|---|---|---|---|---|---|
| | Precision | Recall | F1 | Pearson | AC2 | Kappa |
| *Proprietary models* | | | | | | |
| claude-sonnet-3.7 | 0.350 | 0.622 | **0.448** | 0.380 | 0.339 | 0.355 |
| claude-sonnet-4 | 0.291 | 0.495 | 0.367 | 0.360 | 0.308 | 0.333 |
| gpt-4o | **0.417** | 0.393 | 0.405 | **0.389** | 0.359 | 0.362 |
| gpt-5 | 0.370 | 0.543 | 0.441 | 0.327 | 0.300 | 0.310 |
| gemini-2.5-flash | 0.324 | **0.678** | 0.439 | 0.369 | 0.296 | 0.318 |
| *Open-sourced models* | | | | | | |
| qwen3-235b-a22b | 0.351 | **0.576** | **0.436** | 0.336 | 0.161 | 0.243 |
| llama-3.3 | **0.358** | 0.424 | 0.388 | 0.004 | **0.322** | **0.316** |
| deepseek-v3 | 0.328 | 0.459 | 0.382 | **0.361** | 0.278 | 0.310 |

## F.2 QUALITATIVE RESULTS

We present a thorough qualitative human-model disagreement analysis on REVIEWSCORE evaluation in Figure 9–14. For each review point type, we demonstrate two representative examples which include model-based scores and corresponding justifications using `claude-sonnet-3.7` and `claude-sonnet-4`.

---

**Input: Claim (Review Point) & Paper**

*Paper*:
Towards Continual Knowledge Learning of Language Models (Jang et al., 2022)

---

*Claim (Review Point)*:
The experimental findings in this paper are somewhat trivial.

---

**Human-annotated Label (5-point Scale)**

3, 4, 4 (Median: 4)

---

**Output: ClaimScore Evaluation**

*Model*: `claude-sonnet-3.7`

*Score*: **2**

*Human-model Disagreement Type*: Value judgment

*Justification*: The proposition states that the experimental findings in this paper are somewhat trivial. To evaluate this claim, I need to assess the novelty, significance, and depth of the experimental findings presented in the paper.
Key experimental findings in the paper include:
. . .
These findings are not trivial because:
- They establish a new problem formulation and benchmark for an important real-world challenge
- They provide novel insights about how different methods perform in this setting
- They identify key factors affecting knowledge retention and acquisition
- The findings have practical implications for developing ever-changing language models

---

*Model*: `claude-sonnet-4`

*Score*: **2**

*Human-model Disagreement Type*: Value judgment

*Justification*: To evaluate whether the experimental findings are "somewhat trivial", I need to assess the novelty, significance, and non-obviousness of the results presented in the paper.
**Novel contributions and non-trivial findings:**
. . .
**Overall assessment:**
While some high-level expectations (like catastrophic forgetting occurring) are predictable, the specific findings about which methods work best, the quantitative trade-offs, the multi-phase degradation effects, and the detailed analysis of what causes forgetting provide substantial non-trivial insights. The systematic benchmarking and the discovery that traditional CL methods don't transfer well to this domain are significant contributions.

The experimental work is comprehensive, the findings have practical implications for maintaining up-to-date language models, and several results challenge conventional wisdom from traditional continual learning.

Figure 9: Example #1 of ClaimScore evaluation.

---

**Input: Claim (Review Point) & Paper**

*Paper*:
Temperature Schedules for self-supervised contrastive methods on long-tail data (Kukleva et al., 2023)

---

*Claim (Review Point)*:
The improvement is quite limited compared with other long-tailed learning methods.

---

**Human-annotated Label (5-point Scale)**

2, 2, 3 (Median: 2)

---

**Output: ClaimScore Evaluation**

*Model*: `claude-sonnet-3.7`

*Score*: **3**

*Justification*: The proposition states that "The improvement is quite limited compared with other long-tailed learning methods". To evaluate this claim, I need to examine how the paper compares its proposed temperature scheduling (TS) approach with other long-tailed learning methods. Looking at the paper:

. . .

Given that the paper shows consistent improvements over the baselines and SDCLR, but doesn't compare with a broader range of long-tailed learning methods, and the improvements are modest (1-3%), the proposition that "the improvement is quite limited compared with other long-tailed learning methods" is partially supported but not fully substantiated.

---

*Model*: `claude-sonnet-4`

*Score*: **4**

*Human-model Disagreement Type*: Value judgment

*Justification*: To evaluate this proposition, I need to examine the experimental results and compare the improvements achieved by the proposed temperature scheduling (TS) method against other long-tailed learning methods. Looking at the experimental results:

1. **Comparison with SDCLR**: In Table 3, the paper compares SimCLR+TS with SDCLR (a specialized method for long-tailed contrastive learning). The improvements are modest.
2. **Absolute improvements**: The improvements over baseline methods are generally in the range of 1–4% across different metrics and datasets.
3. **Limited baseline comparisons**: The paper primarily compares against standard contrastive learning methods (MoCo, SimCLR) and only one specialized long-tailed method (SDCLR). It doesn't compare against other established long-tailed learning approaches from supervised learning or other self-supervised long-tailed methods.
4. **Scope of evaluation**: The evaluation is limited to relatively small datasets (CIFAR variants and ImageNet-100) and doesn't demonstrate performance on larger-scale long-tailed datasets that are commonly used in long-tailed learning literature.
5. **Method simplicity vs. improvement**: While the method is simple (just scheduling temperature), the improvements are incremental rather than substantial breakthroughs.

The paper acknowledges this limitation implicitly by positioning the method as "simple yet effective" and emphasizing the zero additional computational cost, rather than claiming state-of-the-art performance improvements.

Figure 10: Example #2 of ClaimScore evaluation.

---

**Input: Premise of Argument (Review Point) & Paper**

*Paper*:
Automatic Chain of Thought Prompting in Large Language Models (Zhang et al., 2023)

-----------------------------------------------------------------------------------------------

*Argument (Review Point)*:
Firstly, when mentioning that the method automatically finds prompts using chain of thoughts (Auto_CoT) approach, this does not imply finding the best samples from the dataset and using the same method as before. In other words, the proposed method does not find the best chain of thought prompts as claimed but rather believes that "let's do this step by step" is the best zero-shot-CoT prompting strategy and the method merely uses that with different prompt samples. The title and the abstract are highly misleading in that way.

-----------------------------------------------------------------------------------------------

*Premise*:
The method uses "let's do this step by step" with different prompt samples.

---

**Human-annotated Label (5-point Scale)**

5, 5, 5 (Median: 5)

---

**Output: ClaimScore Evaluation**

*Model*: `claude-sonnet-3.7`

*Score*: **5**

*Justification*: The premise states that the Auto-CoT method uses "let's do this step by step" with different prompt samples. I need to check if this is accurate based on the paper. Looking at the paper, in the "DEMONSTRATION SAMPLING" section, the author describes how they construct demonstrations:

"A prompted input is formulated as: [Q: q(i)j. A: [P]], where [P] is a single prompt 'Let's think step to step'. This formed input is fed into an LLM using Zero-Shot-CoT (Kojima et al., 2022) to output the reasoning chain consisting of the rationale r(i)j and the extracted answer a(i)j."

. . .

So the premise is accurate - the Auto-CoT method does use "let's think step by step" (or a very similar variant "Let's think step to step") with different prompt samples selected through their clustering and sampling approach.

-----------------------------------------------------------------------------------------------

*Model*: `claude-sonnet-4`

*Score*: **2**

*Human-model Disagreement Type*: Overthinking

*Justification*: To judge the factuality of this premise, I need to examine what the Auto-CoT method actually does according to the paper. From the paper, I can see that:

1. Auto-CoT consists of two main stages: (i) question clustering and (ii) demonstration sampling
2. In the demonstration sampling stage, the method uses Zero-Shot-CoT to generate reasoning chains for selected questions
3. The paper explicitly states that Zero-Shot-CoT uses the prompt "Let's think step by step" (not "let's do this step by step")

. . .

So the premise is mostly accurate in describing the approach (using Zero-Shot-CoT with different samples), but contains a factual error in the exact wording of the prompt. The paper consistently uses "Let's think step by step" rather than "let's do this step by step."

Figure 11: Example #1 of ArgScore evaluation.

---

**Input: Premise of Argument (Review Point) & Paper**

*Paper*:
Automatic Chain of Thought Prompting in Large Language Models (Zhang et al., 2023)

---

*Argument (Review Point)*:
Secondly, this is very obvious to anyone that the prompts must be representative of the test samples (as much as possible). This work takes the dataset, clusters them, finds the most representative sample from each cluster, and uses that as prompts. There is nothing extra-ordinary about this approach and is merely a small improvement over the randomly sampling strategy. This can also be seen from a marginal improvement in the results over a random sampling strategy.

---

*Premise*:
The approach is obvious (i.e., prompts must be representative of test samples).

---

**Human-annotated Label (5-point Scale)**

4, 4, 5 (Median: 4)

---

**Output: ClaimScore Evaluation**

*Model*: `claude-sonnet-3.7`

*Score*: **2**

*Human-model Disagreement Type*: Misunderstanding / incorrect reasoning

*Justification*: I'll examine whether the paper itself acknowledges or implies that prompts should be representative of test samples as an obvious approach:
. . .
The paper's main contribution is showing that diversity in demonstrations is important, not just representativeness. The paper states: "diversity-based clustering may mitigate misleading by similarity" and "diversity of demonstration questions is the key."

This suggests that the premise oversimplifies the paper's findings. The paper doesn't claim that prompts must simply be representative – it argues for a balance of representativeness and diversity.

---

*Model*: `claude-sonnet-4`

*Score*: **4**

*Justification*: To judge the factuality, I need to determine whether it's true that "prompts must be representative of test samples" is an obvious principle. This is a general statement about machine learning methodology and prompt engineering principles that would be known to AI practitioners before 2023. The idea that training/demonstration examples should be representative of test data is a fundamental principle in machine learning that predates this specific paper. This principle is well-established in the field and doesn't require knowledge of this specific paper to evaluate its truth.

The paper does implement this principle through clustering and selecting representative samples, but the general principle itself is established knowledge in the field.

Figure 12: Example #2 of ArgScore evaluation.

---

**Input: Question (Review Point) & Paper**

*Paper*:
ReAct: Synergizing Reasoning and Acting in Language Models (Yao et al., 2023)

---

*Question (Review Point)*:
Is there a way to add self-consistency to ReAct, such that it samples multiple reasoning traces and actions and uses the majority action, for example? SC seems extremely effective for Chain of Thought tasks, and it seems like something similar could work here.

---

**Human-annotated Label (5-point Scale)**

2, 2, 5 (Median: 2)

---

**Output: ClaimScore Evaluation**

*Model*: `claude-sonnet-3.7`

*Score*: **4**

*Human-model Disagreement Type*: Minority score prediction

*Justification*: The question asks about adding self-consistency to ReAct, specifically whether it's possible to sample multiple reasoning traces and actions and use the majority action, similar to how self-consistency works for Chain of Thought tasks. Let me check if the paper discusses this concept or something similar:

. . .

However, the paper does not explicitly discuss applying the self-consistency technique directly to ReAct itself (i.e., sampling multiple ReAct trajectories and taking the majority action). It only discusses combining ReAct with CoT-SC as separate methods, not applying the self-consistency sampling technique to ReAct's own trajectories.

---

*Model*: `claude-sonnet-4`

*Score*: **4**

*Human-model Disagreement Type*: Minority score prediction

*Justification*: The question asks about adding self-consistency to ReAct, similar to how it's used in Chain of Thought (CoT) tasks. Let me examine what the paper says about self-consistency and whether it discusses or implements this for ReAct. Looking through the paper, I can see that:

. . .

However, the question specifically asks about applying self-consistency directly to ReAct itself – sampling multiple ReAct reasoning traces and actions and using majority voting. While the paper discusses CoT-SC and hybrid approaches that combine ReAct with CoT-SC, it doesn't appear to directly implement or discuss a "ReAct-SC" approach where multiple ReAct trajectories are sampled and majority voting is applied to the ReAct outputs themselves.

Figure 13: Example #1 of QScore evaluation.

---

**Input: Question (Review Point) & Paper**

*Paper*:
Large Language Models Can Self-improve (Huang et al., 2023)

---

*Question (Review Point)*:
Can you check if baseline self-consistency with more examples can match LMSI or not?

---

**Human-annotated Label (5-point Scale)**

4, 5, 5 (Median: 5)

---

**Output: ClaimScore Evaluation**

*Model*: `claude-sonnet-3.7`

*Score*: **2**

*Human-model Disagreement Type*: Misunderstanding / incorrect reasoning

*Justification*: The paper includes a hyperparameter study in Figure 4(b) that shows the effect of varying the number of sampled reasoning paths for self-consistency, both with and without LMSI. This directly addresses whether increasing the number of examples in baseline self-consistency can match LMSI performance.

---

*Model*: `claude-sonnet-4`

*Score*: **4**

*Justification*: Looking at the experimental setup and results:
 - The paper uses m=32 reasoning paths for generating training data
 - Figure 4(b) shows accuracy vs number of sampled reasoning paths for both w/o LMSI and LMSI
 - The experiments seem to use consistent numbers of paths for comparison

The paper doesn't systematically explore whether dramatically increasing the number of paths in baseline self-consistency (e.g., to 100+ paths) could match the LMSI performance. While Figure 4(b) shows diminishing returns after 15–20 paths, it doesn't explore very high numbers of paths that might be computationally equivalent to the LMSI training process.

Figure 14: Example #2 of QScore evaluation.

## G    MODEL DETAILS

We measure reliability of automatic REVIEWSCORE evaluation with eight current state-of-the-art LLMs, including five proprietary models and three open-sourced models. For proprietary models, we use Anthropic's `claude-3-7-sonnet-20250219` (Anthropic, 2025a) and `claude-sonnet-4-20250514` (Anthropic, 2025b), OpenAI's `gpt-4o-2024-08-06` (OpenAI, 2024) and `gpt-5-2025-08-07` (OpenAI, 2025), and Google's `gemini-2.5-flash` (Gemini Team, Google, 2025). Since the REVIEWSCORE evaluation does not require longer chain of thought, we exclude reasoning models. Following this rule, we use Anthropic's claude models without thinking modes and OpenAI's gpt-5 model with minimal reasoning effort. For open-sourced models, we use Alibaba's `qwen3-235b-a22b-2507` (Yang et al., 2025) with 235B model parameters, DeepSeek's `deepseek-v3-0324` (DeepSeek-AI et al., 2024) with 671B model parameters and Meta's `llama-3.3-70b-instruct` (AI@Meta, 2024) with 70B model parameters. We call the models by their APIs.

## H    PROMPTS

We list every prompt used for automatic REVIEWSCORE evaluation (Section 4), automatic review point type classification (used in preprocessing stage of the dataset construction in Section 3.3), and automatic argument reconstruction (Section 3.2). Figure 15 – Figure 20 indicate prompts for automatic REVIEWSCORE evaluation and related score rubrics. Figure 21 and Figure 22 indicate prompts for automatic review point type classification, where the type is one of claim, argument, and question. Figure 23 – Figure 29 indicate prompts for automatic argument reconstruction. In the prompts, capital letters with double square brackets (i.e., [[XYZ]]) would be replaced by the corresponding material, and words with a curly bracket (i.e., {abc}) would be replaced by the corresponding score rubric (Figure 15 – Figure 17) or a python code snippet in Figure 26.

---

**Unanswerability Score Rubric**

1: Definitely Answerable by the paper
2: Probably Answerable by the paper
3: No Verdict
4: Probably Unanswerable by the paper
5: Definitely Unanswerable by the paper

Figure 15: Score rubric for evaluating *unswerability*.

---

**Factuality Score Rubric**

1: Definitely False
2: Probably False
3: No Verdict
4: Probably True
5: Definitely True

Figure 16: Score rubric for evaluating *factuality*.

---

**Untrivialness Score Rubric**

0: Definitely Trivial
1: Probably Trivial
2: Definitely Not Trivial

Figure 17: Score rubric for evaluating *untrivialness*.

---

**Prompt for evaluating QScore (*unanswerability of questions*)**

# Paper
[[PAPER]]

# Question
[[QUESTION]]

You are given a paper submitted to an AI conference and a question regarding the paper.

Judge if the question is answerable or not by the paper. You should scale 1-5 to indicate unanswerability as follows.
{Unanswerability Score Rubric}

If you score the question 1 or 2, then indicate which knowledge source you have grounded to (i.e., indicate corresponding section(s) and verbatim sentence(s)) and answer the question.

Your output should be the following.

### Reasoning
[think step-by-step]

### Unanswerability
[1 or 2 or 3 or 4 or 5]

### Source
[corresponding section(s) and verbatim sentence(s) if the score is 1 or 2, otherwise None]

### Answer
[answer the question if the score is 1 or 2, otherwise None]

---

Figure 18: Prompt used for evaluating QScore in a 5-point scale.

---

**Prompt for evaluating ClaimScore or WScore (*factuality of weaknesses*)**

# Paper
[[PAPER]]

# Weakness
[[WEAKNESS]]

You are given a paper submitted to an AI conference and a weakness regarding the paper.

Judge if the weakness is true or not based on the paper. You should scale 1-5 to indicate factuality as follows.
{Factuality Score Rubric}

Your output should be the following.

### Reasoning
[think step-by-step]

### Factuality
[1 or 2 or 3 or 4 or 5]

---

Figure 19: Prompt used for evaluating ClaimScore or WScore in a 5-point scale.

---

**Prompt for evaluating ArgScore (*factuality of premises*)**

# Paper
[[PAPER]]

# Weakness
[[WEAKNESS]]

# Premise
[[PREMISE]]

You are given a paper submitted to an AI conference, a weakness of the paper, and one of premises of the weakness. Your task is to judge the factuality and untrivialness of the given premise.

First, judge the factuality of the premise. To do that, choose an appropriate knowledge source from:
    1. given paper
    2. annotator knowledge before the year [[YEAR]] (more precisely, before the paper is publicized)
    3. other paper(s),
and then judge the factuality of the premise based on the knowledge source. You should scale 1-5 to indicate factuality as follows.
{Factuality Score Rubric}

Here are guidelines you should follow:
 - Main purpose is to distinguish *given_paper* and *annotator_knowledge*.
 - Select *other_papers* only if the premise refers a specific paper.
 - Note that you should separate judging the factuality of the premise from understanding the semantics of the premise. It does not matter whether *given_paper* is needed or not to understand the semantics of the premise. The knowledge source is *given_paper* only if *given_paper* is needed to judge the factuality of the premise, otherwise, the knowledge source is *annotator_knowledge*. For example, although you need the paper's context in order to understand what the premise means, if you do not need the paper's knowledge to judge if the premise is true or not (e.g., logical assessment), then you should choose *annotator_knowledge* as a knowledge source and judge the factuality accordingly.
 - For premises that are conditionals (If A then B), you should presume that the antecedent (A) is always true even if the antecedent does not align with the paper's knowledge. (Because the antecedent is always true, the knowledge source should only be determined while judging the factuality of the consequent.) Then, choose an appropriate knowledge source to judge if the consequent (B) is true or not and judge the factuality accordingly.

Next, decide whether the premise is trivially true or not based on the common knowledge of CS/AI-majoring undergrad students before the year [[YEAR]] (more precisely, before the paper is publicized). For premises that are conditionals (If A then B), you must assume that the antecedent (A) is true and judge if the consequent (B) is trivially true or not. You should scale the score to 0-2 as follows.
{Untrivialness Score Rubric}

Here are guidelines you should follow:
- If the knowledge source is *given_paper* or *other_papers*, then untrivialness should always be 2 unless the premise factuality could also be determined by *annotator_knowledge*.
- If the knowledge source is *annotator_knowledge*, then untrivialness could be 0-2.

Your output should be formatted as below.

### Reasoning
[think step-by-step]

### Source
[given_paper or annotator_knowledge or other_papers]

### Factuality
[1 or 2 or 3 or 4 or 5]

### Untrivialness
[0 or 1 or 2]

---

Figure 20: Prompt used for evaluating ArgScore in a 5-point scale.

---

**Prompt for classifying review point types (`is_argument`)**

---

\# Review Point
[[REVIEW_POINT]]

You are given an AI conference review point. Is this an argument or not? Your response should follow the format below.

\#\#\# Reasoning
[think step-by-step]

\#\#\# Response
[Yes or No]

---

Figure 21: Prompt used for deciding if a review point is an argument or not (is_argument).

---

**Prompt for classifying review point types (`is_question`)**

---

\# Review Point
[[REVIEW_POINT]]

You are given an AI conference review point. Decide if this is a question or a simple claim. Your response should follow the format below.

\#\#\# Reasoning
[think step-by-step]

\#\#\# Response
[Question or Claim]

---

Figure 22: Prompt used for deciding if a review point is a question or a claim (is_question).

---

**Prompt for Argument Reconstruction (`extract_verbatim_conclusion_reason`)**

---

\#\#\# Review Point
[[REVIEW_POINT]]

Given an AI conference review point, consider it as an argument, and then returns its verbatim conjecture in the source text and verbatim reason statements of that conjecture in the source text. The output format should be as following.

\#\#\# Conjecture
[main conjecture in the review point]

\#\#\# Supporting Reasons
[list of supporting reasons for the conjecture]

---

Figure 23: Prompt used for extracting verbatim conjecture and reason statements in an argument (extract_verbatim_conclusion_reason).

---

**Prompt for Argument Reconstruction (`argument_reconstruction`)**

# Paper
[[PAPER]]

# Review Point

## Conclusion
[[CONCLUSION]]

## Explicit reasons
[[REASONS]]

You are given a paper submitted to an AI conference and a review point by a peer reviewer. A review point consists of a conclusion and its explicit reasons. Reconstruct an argument (i.e., a review) with premise-conclusion structure where premises deductively imply the conclusion. The reconstructed argument should be deductively valid, using formal logical patterns like modus ponens (e.g., Premise1: A, Premise2: If A then B, Conclusion: B). Add implicit premises and intermediate conclusions if needed.

Your output should composed of two parts, argument reconstruction and its formalization. In the first part, list premises, intermediate conclusions, and conclusion, and indicate their logical connection (i.e., which propositions deductively implies which). In the second part, first define variables and/or predicates, then formalize premises, intermediate conclusions, and a conclusion, and then generate a deductive proof. The output format should be as following.

# Argument Reconstruction

## Premises
[list of explicit and implicit premises]

## Intermediate Conclusions
[list of intermediate conclusions (if intermediate conclusions are not needed, then write "None".)]

## Conclusion
[a conclusion]

## Logical Connections
[list of logical connections]

# Formalized Argument

## Defined Variables/Predicates
[definition of each variable and/or predicate]

## Formalized Premises
[formalization of premises using definition]

## Formalized Intermediate Conclusions
[formalization of intermediate conclusions using definition (if intermediate conclusions are not needed, then write "None".)]

## Formalized Conclusion
[formalization of conclusion using definition]

## Deductive Proof
[deductive proof using formalized premises]

Figure 24: Prompt used for reconstructing an argument (`argument_reconstruction`).

---

**Prompt for Argument Reconstruction (`streamlining`)**

## Defined Variables/Predicates
[[DEFINITION]]

## Formalized Premises
[[PREMISES]]

## Formalized Conclusion
[[CONCLUSION]]

## Deductive Proof
[[PROOF]]

First, determine necessary formalized premises for the given deductive proof. This includes:
 1. Add any missing formalized premises that are necessary to prove conclusion but cannot be dervied from the formalized premises.
 2. Remove any unnecessary formalized premises that are not necessary to prove conclusion but present in the formalized premises.
You should format these premises into a python dictionary where keys and values are python strings.

Second, write a python program using z3 that inputs the necessary formalized premises and formalized conclusion and outputs:
 1. Their validity, formatted as a python string of either "valid" or "invalid".
 2. A smallest subset of necessary formalized premises to prove the formalized conclusion, formatted as a python list of keys of the python dictionary of the necessary formalized premises.
You should therefore print two things (a python string and a python list) separately. Please use the below python code snippet.
{Code snippet for checking validity}

Third, return the final formalized conclusion that is used in the python program in step 2.

Lastly, judge whether the formal proof using the necessary formalized premises (in step 1) and the final formalized conclusion (in step 3) is circular or not. If there is a single necessary formalized premise that is the same as the final formalized conclusion, then return N/A.

Your response should be as following.

### Necessary Formalized Premises
``` python
{
    "[Symbol of a premise #1]": "[Formalization of a premise #1]",
    "[Symbol of a premise #2]": "[Formalization of a premise #2]",
    ...
}
```

### Python Program
``` python
[a python program]
```

### Final Formalized Conclusion
[Formalized conclusion in the python program]

### Proof Circularity
[Yes or No or N/A]

Figure 25: Prompt used for streamlining formalized reconstruction (`streamlining`).

**Code snippet for checking validity**

```python
from z3 import *
import itertools

###################################
### Write down your code here ###
###################################

# Check validity of the argument
def check_validity(premises_dict, conclusion):
    s = Solver()
    s.add(list(premises_dict.values()))
    s.add(Not(conclusion))
    if s.check() == unsat:
        return "valid"
    else:
        return "invalid"

# Find minimal set of premises
def find_minimal_premises(premises_dict, conclusion):
    for subset_size in range(1, len(premises_dict) + 1):
        for subset in itertools.combinations(premises_dict.keys(), subset_size):
            subset_premises = [premises_dict[key] for key in subset]
            if check_validity(subset_premises, conclusion) == "valid":
                return list(subset)
    return list(premises_dict.keys())

validity = check_validity(premises, conclusion)
print(validity)
minimal_premises = find_minimal_premises(premises, conclusion)
print(minimal_premises)
```

Figure 26: Python code snippet for evaluating validity of reconstruction.

**Prompt for Argument Reconstruction (`program_debugging`)**

When I execute the python program, I got the following error:
[[ERROR]]

Fix the error and generate a revised python program. Your response should be as following.

### Reasoning
[explain why and how to fix the program]

### Revised Python Program
```python
[a python program]
```

Figure 27: Prompt used for debugging python programs that evaluate validity of reconstruction (`program_debugging`).

---

**Prompt for Argument Reconstruction (`deformalization`)**

## Defined Variables/Predicates
[[DEFINITION]]

## Formalized Premises
[[PREMISES]]

## Formalized Conclusion
[[CONCLUSION]]

Given definitions of variables and/or predicates, generate natural language (NL) descriptions of formalized premises and conclusion. Your response should be as following.

### NL Premises
[list of premises in natural language]

### NL Conclusion
[conclusion in natural language]

---

Figure 28: Prompt used for translating FOL formulas with keys (i.e., defined variables/predicates) to NL reconstructed arguments (`deformalization`).

---

**Prompt for Argument Reconstruction (`check_faithfulness`)**

# Argument
[[ARGUMENT]]

# Argument Reconstruction

## Premises
[[PREMISES]]

## Conclusion
[[CONCLUSION]]

For an argument, its reconstruction as a premise-conclusion structure is given. Your task is to judge whether the construction is faithful or not. You should judge the faithfulness according to the following two criteria:
 - **Accuracy & Charity.** The reconstruction should keep the author's intended meaning while eliminating irrelevancies—i.e., obey the principle of charity and prefer the strongest sensible reading of ambiguous passages.
 - **Completeness.** All explicit premises, the main conclusion and any indispensable implicit premises must be included.

The output format should be as following.

# Reasoning
[Explain step-by-step]

# Faithfulness
[Yes or No]

---

Figure 29: Prompt used for evaluating faithfulness of reconstruction (`check_faithfulness`).

# I ADDITIONAL COMPARISON WITH PRIOR WORKS

## I.1 DATASET STATISTICS

We compare dataset statistics of REVIEWSCORE with other datasets in Table 14. "-" indicates the information is not explicitly addressed in the paper. We indicate the statistics of human-annotated reviews. In terms of number of papers, our work has a similar size with Ou et al. (2025) and has a smaller size than Shin et al. (2025) and Du et al. (2024). However, in terms of number of review points, which is the primary measure of review data size, our work has a larger data size than Ou et al. (2025), Shin et al. (2025), and Purkayastha et al. (2025) since they filter out reviews that does not fit to their problem setting (We use all review points as targets for evaluation). Furthermore, while other works focus on evaluating at a level of review points, we also evaluate at a level of premises of review points, which are more fine-grained. As a result, our human experts annotate total 1,748 premises of review points. To guarantee annotation reliability, each instance is labeled by three human experts. This design choice intentionally prioritizes quality over scale, resulting in a smaller dataset under limited human annotation capacity. Furthermore, other works mostly evaluate reviews of NLP papers, whereas our work includes other domains such as CV and Time Series.

| | # Papers | # RP | # Prem. | Ann./Inst. | Domains | Venues |
|---|---|---|---|---|---|---|
| Ou et al. (2025) | 41 | 168 | N/A | – | NLP | NeurIPS 2023–2024 |
| Shin et al. (2025) | 68 | 327 | N/A | 2 | – | ICLR 2021–2024 |
| Purkayastha et al. (2025) | – | 500 | N/A | 2 | NLP | ARR 2022 |
| Du et al. (2024) | 100 | N/A | N/A | 2 | NLP | ICLR/NeurIPS 2020–2023 |
| REVIEWSCORE (Ours) | 40 | **657** | **1,748** | 3 | CV, NLP, Time Series | ICLR 2021–2023 |

Table 14: Comparison of dataset statistics. RP = Review Points. Prem. = Premises.

## I.2 REVIEW EVALUATION METHODS

We compare review evaluation methods of REVIEWSCORE with other works in Table 15. To be specific, we compare evaluation targets and units, and review quality criteria. These four criteria are adopted from Goldberg et al. (2025) and other review evaluation works. While most of previous works focus on evaluating weaknesses, we additionally focus on evaluating questions. As a result, we are the only work that consider not only factuality of weaknesses but also unanswerability of questions. Most of the works including REVIEWSCORE evaluate review points, and we additionally evaluate premises of review points, which we define factuality at a level of premises. In contrast, Du et al. (2024) evaluates factuality of each sentence. However, sentence-level factuality is less reliable than premise-level factuality since human reviews could be repetitive, hide implicit premises, contain rhetoric, and sometimes change a stance in a single review. Premise-level factuality is robust to all of these effects. Guo et al. (2023), Sadallah et al. (2025), and Ou et al. (2025) consider groundedness of reviews to the paper, but do not consider factuality and leave it as a future work. Our work and Du et al. (2024) both considers groundedness and factuality. However, since we are focusing of detecting misinformation in reviews, we do not evaluate constructiveness of reviews. We also think that evaluating constructiveness is more subjective and value-laden than other criteria, which requires more careful human annotations.

| | Targets | Units | Groundedness | Factuality | (Un)Answerability | Constructiveness |
|---|---|---|---|---|---|---|
| Guo et al. (2023) | S, W | R | ✓ | ✗ | ✗ | ✗ |
| Sadallah et al. (2025) | W | RP | ✓ | ✗ | ✗ | ✓ |
| Ou et al. (2025) | W | RP | ✓ | ✗ | ✗ | ✗ |
| Shin et al. (2025) | S, W | RP | ✗ | ✗ | ✗ | ✗ |
| Purkayastha et al. (2025) | W | RP | ✗ | ✗ | ✗ | ✗ |
| Du et al. (2024) | W | RS | ✓ | ✓ | ✗ | ✓ |
| REVIEWSCORE (Ours) | W, Q | RP & Prem. | ✓ | ✓ | ✓ | ✗ |

Table 15: Comparison of review evaluation methods. S = Strengths. W = Weaknesses. Q = Questions. R = Reviews, RP = Review Points. RS = Review Sentences. Prem. = Premises.

