# OpenReview forum: "ReviewScore: Misinformed Peer Review Detection with Large Language Models"
_ICLR.cc/2026/Conference — Submitted to ICLR 2026_

### Official Review · Reviewer_ywZt · 2025-10-30

**Soundness:** 3
**Presentation:** 3
**Contribution:** 3
**Rating:** 4
**Confidence:** 3

**Summary:**

This paper addresses the degrading quality of peer review in AI conferences, a problem driven by the explosive growth in paper submissions. The authors aim to reliably detect low-quality review by defining "misinformed review points" as either reviewer "questions" that are already answerable by the paper or "weaknesses" that are incorrect or based on incorrect premises. The primary contribution is ReviewScore, a new evaluation criterion to automatically identify these misinformed points. The paper introduces base ReviewScore for foundational scoring and a more effective advanced ReviewScore, which breaks arguments down to evaluate the factuality of each individual premise. To enable this, the authors also propose an automatic argument reconstruction engine that uses LLMs to extract all explicit and implicit premises from a review. To validate this automated system, the authors built a human expert-annotated ReviewScore dataset and measured the performance of eight state-of-the-art LLMs against it. The results showed moderate human-model agreement and confirmed that evaluating premise-level factuality is significantly more reliable than evaluating weakness-level factuality.

**Strengths:**

* This paper introduces a refined approach to reviewing AI conference papers by providing a more detailed examination of each review point.
* The paper proposes the ADVANCED REVIEWSCORE, which evaluates the factuality of individual premises within arguments, proving to be more effective than simply evaluating weaknesses at a high level.

**Weaknesses:**

1. While the paper introduces the REVIEWSCORE system to detect misinformed review points, its classification of errors into two categories: questions and weaknesses, appears too simplistic. There are likely more nuanced types of errors in peer reviews that this system does not currently address.
2. Although the paper presents REVIEWSCORE to detecting misinformed review points, it fails to adequately compare its method with existing peer review evaluation systems.
3. The paper’s experimental evaluation relies on a small and homogeneous dataset consisting of 40 ICLR submissions. While this serves as a proof of concept, the limited size and scope of the dataset restrict its generalizability. The dataset's lack of diversity in terms of conferences and academic domains may not reflect the wide range of review styles and subjects encountered across various fields.

**Questions:**

1. What is the rationale behind limiting the classification to just these two categories? Are there other types of misinformed review points, such as misunderstandings of methodologies, misinterpretation of results, or reviewer biases, that could be considered?
2. Could you provide a more in-depth comparative analysis to highlight the strengths and limitations of REVIEWSCORE in relation to other state-of-the-art review quality assessment tools? How does REVIEWSCORE perform in comparison to other AI-based approaches for identifying misinformed review points?
3. How do you plan to address the generalizability of your findings given the limited scope of the dataset? Additionally, there could be biases introduced during the human annotation process. Could you expand the dataset to include a wider range of conferences and academic domains to ensure robustness? Also, could the inclusion of external data sources (e.g., citation counts, feedback from future users) strengthen the accuracy and completeness of the dataset?

---

> ### Author Response · Authors · 2025-11-21
> **Rebuttal by Authors (1/4)**
>
> We thank the reviewer for the constructive and detailed comments. **Q** denotes a Question.
>
> **Q1. What is the rationale behind limiting the classification to just these two categories? Are there other types of misinformed review points, such as misunderstandings of methodologies, misinterpretation of results, or reviewer biases?**
>
> A. First, as described in Introduction (line 40-47), the criteria of misinformed review points are decided by human experts' independent analyses on a small subset of ICLR reviews and their follow-up discussion about trustworthy criteria to detect low-quality review points.
> Also, in Appendix B, we further describe the detailed process of review quality criteria discussion. The following is the relevant paragraph that could address the reviewer's concern.
>
> > Then, we categorize these justifications in order to find common features of low-quality review points.
> There are five common types: (1) questions that can be already addressed by the paper, (2) comments that reflect a misunderstanding of the paper, (3) out-of-scope remarks, (4) observations pointing out minor details, and (5) unclear points.
> However, for the last three types, we observe that majority of review points are agreed or argued by a single human annotator. In other words, given a review point, only a single human argues it is out-of-scope, whereas other two humans argue it is within scope and could be a potential drawback of the submitted paper. Following this pattern, a single human argues a review point is addressing minor details but others argue it is a major point, and a single human argues a review point is unclear whereas others do not agree with this.
> In contrast, the first two types of review points (i.e., answerable questions and misunderstood comments) are mostly agreed by two or more human annotators, indicating that these two types have more objective and trustworthy criteria for detecting low-quality reviews. Based on these observations, we focus on evaluating review points based on the first two criteria.
>
> There are more details in Appendix B.
>
> Second, other types of misinformed review points that the reviewer mentions are covered by our two proposed criteria.
> If review points are misunderstandings of methodologies or misinterpretation of results, then those are included in weaknesses that contain incorrect premises regarding the paper.
> For reviewer biases, if a reviewer bias manifests as a factual error, then it becomes an incorrect premise and is flagged. However, if a bias is purely preferential (e.g., “I dislike synthetic pretraining”), it won't be flagged since there is no falsifiable error. We want to emphasize that ReviewScore focuses on evidence-checkable misinformation.

---

> > ### Author Response · Authors · 2025-11-21
> > **Rebuttal by Authors (2/4)**
> >
> > **Q2. How does REVIEWSCORE perform in comparison to other AI-based approaches for identifying misinformed review points?**
> >
> > A. In the first paragraph of Introduction (line 33-39), we briefly mention that previous works on review evaluation have a trade-off between applicability to reviews and specificity of rubrics, and to resolve this issue, we introduce ReviewScore.
> > In the first the paragraph of Related Works, we further describe how each previous work is limited and how we resolve these issues in our work.
> > To compare our work with previous works more comprehensively, we summarize it as the following table.
> >
> > |                               | Evaluation Targets    | Evaluation Units                 | Groundedness  | Factuality  | (Un)Answerability  | Constructiveness  |
> > | ----------------------------- | --------------------- | -------------------------------- | ------------  | ----------  | -----------------  | ----------------  |
> > | **Guo et al. (2023)**         | strengths, weaknesses | reviews                          | ✅            | ❌          | ❌                 | ❌                |
> > | **Sadallah et al. (2025)**    | weaknesses            | review points                    | ✅            | ❌          | ❌                 | ✅                |
> > | **Ou et al. (2025)**          | weaknesses            | review points                    | ✅            | ❌          | ❌                 | ❌                |
> > | **Shin et al. (2025)**        | strengths, weaknesses | review points                    | ❌            | ❌          | ❌                 | ❌                |
> > | **Purkayastha et al. (2025)** | weaknesses            | review points                    | ❌            | ❌          | ❌                 | ❌                |
> > | **Du et al. (2024)**          | weaknesses            | review sentences                 | ✅            | ✅          | ❌                 | ✅                |
> > | **ReviewScore**               | weaknesses, questions | review points and their premises | ✅            | ✅          | ✅                 | ❌                |
> >
> > In this table, we compare evaluation targets and units, and review quality criteria. These four criteria are adopted from Goldberg et al. (2025) and other review evaluation works.
> > While most of previous works focus on evaluating weaknesses, we additionally focus on evaluating questions. As a result, we are the only work that consider not only factuality of weaknesses but also unanswerability of questions.
> > Most of the works including ReviewScore evaluate review points, and we additionally evaluate premises of review points, which we define factuality at a level of premises. In contrast, Du et al. (2024) evaluates factuality of each sentence. We want to highlight that sentence-level factuality is less reliable than premise-level factuality since human reviews could be repetitive, hide implicit premises, contain rhetoric, and sometimes change a stance in a single review. Premise-level factuality is robust to all of these effects.
> > Guo et al. (2023), Sadallah et al. (2025), and Ou et al. (2025) consider groundedness of reviews to the paper, but do not consider factuality and leave it as a future work. Our work and Du et al. (2024) both considers groundedness and factuality.
> > However, since we are focusing of detecting misinformation in reviews, we do not evaluate constructiveness of reviews. We also think that evaluating constructiveness is more subjective and value-laden than other criteria, which requires more careful human annotations.
> > We add this comparison in Appendix I.2.

---

> ### Author Response · Authors · 2025-11-21
> **Rebuttal by Authors (3/4)**
>
> **Q3-1. How do you plan to address the generalizability of your findings given the limited size and scope of the dataset? Could you expand the dataset to include a wider range of conferences and academic domains to ensure robustness?**
>
> A. First, to address the limited size of the dataset, we include the response to **W1** of the reviewer **ht6d** in the following.
>
> > Compared to other works on review evaluation, we want to highlight that evaluating LLMs on reviews of 40 ICLR submissions is a plausible choice. A detailed comparison is shown in the table below.
> >
> > | Dataset                       | # Papers | # Review Points | # Premises | # Annotators / Instance | # Total Annotators | Domains              | Venues                 |
> > | ------------------            | -------- | --------------- | ---------- | ----------------------- | ------------------ | -------------------- | ---------------------- |
> > | **Ou et al. (2025)**          | 41       | 168             | N/A        | –                       | 6                  | NLP                  | NeurIPS 2023–2024      |
> > | **Shin et al. (2025)**        | 68       | 327             | N/A        | 2                       | 2                  | –                    | ICLR 2021–2024         |
> > | **Purkayastha et al. (2025)** | –        | 500             | N/A        | 2                       | 4+                 | NLP                  | ARR 2022               |
> > | **Du et al. (2024)**          | 100      | N/A             | N/A        | 2                       | 40                 | NLP                  | ICLR/NeurIPS 2020–2023 |
> > | **Our's**                     | 40       | **657**         | **1,748**  | 3                       | 15                 | CV, NLP, Time Series | ICLR 2021–2023         |
> >
> > In the table, "-" indicates the information is not explicitly addressed in the paper. We indicate the statistics of human-annotated reviews.
> In terms of number of papers, our work has a similar size with Ou et al. (2025) and has a smaller size than Shin et al. (2025) and Du et al. (2024).
> However, in terms of number of review points, which is the primary measure of review data size, our work has a larger data size than Ou et al. (2025), Shin et al. (2025), and Purkayastha et al. (2025) since they filter out reviews that does not fit to their problem setting (We use all review points as targets for evaluation). Furthermore, while other works focus on evaluating at a level of review points, we also evaluate at a level of premises of review points, which are more fine-grained. As a result, our human experts annotate total 1,748 premises of review points.
> To guarantee annotation reliability, each instance is labeled by three human experts. This design choice intentionally prioritizes quality over scale, resulting in a smaller dataset under limited human annotation capacity.
> We also want to emphasize that other works mostly evaluate reviews of NLP papers, whereas our work includes other domains such as CV and Time Series.
> We add this comparison in Appendix I.1.
>
> Second, to address the limited scope of the dataset, we include the response to **Q2** of the reviewer **ht6d** in the following.
>
> > If the reviewer is asking whether a ReviewScore evaluation would generalize on other AI conferences, we perform an extra experiment on reviews from ARR 2022. Due to the limited discussion period, we use a human expert to annotate reviews of a single paper in the NLPeer dataset (Dycke et al., 2023). Following the same setup with our main experiment, claude-sonnet-3.7 achieves 0.385 Kappa score and 0.333 F1 score, which shows similar human-model alignment with our main result. This result shows a potential of ReviewScore to generalize on other AI conferences.
> Furthermore, other works on review evaluation (Ou et al., 2025; Shin et al., 2025; Purkayastha et al., 2025) also target a single AI conference. However, as shown in the table in the response of **W1**, we covered not only NLP domain but also other domains such as CV and Time Series.
> We also want to point out that outside of AI conferences is out of our motivation. Misinformed reviews are problems of current AI conferences, not in other domains, so we want to clarify that these other domains are not our target. However, since ReviewScore is a domain-generalizable evaluation framework, we think that it would work on other academic domains. We leave testing this generalizability as a future work.

---

> > ### Author Response · Authors · 2025-11-21
> > **Rebuttal by Authors (4/4)**
> >
> > **Q3-2. Additionally, there could be biases introduced during the human annotation process.**
> >
> > A. Yes, we agree with that any kinds of bias could be introduced during human annotation. To minimize these biases and maximize trustworthiness, we want to emphasize that we carefully designed the human annotation process.
> > To be specific, as described in line 293-301, we provide human annotators detailed guidelines and an orientation session. Then, three humans who share research interests independently annotate reviews. During annotation, we highly encourage human annotators to communicate with us through a group chat so that we could give them instant responses to their questions and share with all, which builds a global consensus among human annotators.
> > Despite these efforts, as described in Limitations (line 467-472), there could still exist unavoidable biases such as incorrect human annotation due to a significant amount of cognitive load.
> >
> > **Q3-3. Could the inclusion of external data sources (e.g., citation counts, feedback from future users) strengthen the accuracy and completeness of the dataset?**
> >
> > A. Yes, it could strengthen the completeness of the dataset and further enriches human annotations.
> > For citation counts, we could check if the papers with less misinformed review points eventually have higher number of citations. However, since the number of misinformed review points more heavily depends on reviewers' understanding of papers, the citation counts of papers and their number of misinformed review points might not highly correlate with each other. Therefore, we could use citation counts as a complementary validity check rather than new labels.
> > For the future-user feedback, I think this is a great idea. We could make an online platform to collect user feedback or run a pilot study on a real conference to collect feedback from ACs or meta-reviewers. The form of feedback could be a direct correctness rating for each review point, or an indirect helpfulness rating for each paper or review.
> >
> > ---
> > [1] Automatic analysis of substantiation in scientific peer reviews. Guo et al., 2023.
> > [2] The Good, the Bad and the Constructive: Automatically Measuring Peer Review's Utility for Authors. Sadallah et al., 2025.
> > [3] CLAIMCHECK: How Grounded are LLM Critiques of Scientific Papers? Ou et al., 2025.
> > [4] Mind the Blind Spots: A Focus-Level Evaluation Framework for LLM Reviews. Shin et al., 2025.
> > [5] LazyReview A Dataset for Uncovering Lazy Thinking in NLP Peer Reviews. Purkayastha et al., 2025.
> > [6] LLMs assist NLP researchers: Critique paper (meta-) reviewing. Du et al., 2024.

---

> > > ### Comment · Reviewer_ywZt · 2025-11-26
> > > **Thanks for the rebuttal, but concerns on subjectivity and definition remain.**
> > >
> > > I thank the authors for their detailed rebuttal, especially the comprehensive comparison table with prior works and the additional experiment on the ARR dataset.
> > >
> > > However, after carefully reading the rebuttal and other reviewers' comments, I have decided to maintain my original score. My primary concern regarding the subjectivity and the rationale of the classification has not been fully resolved, and in some aspects, reinforced by the discussions.
> > >
> > > - As noted by other reviewers, the moderate inter-annotator agreement (Kappa ~0.35) empirically confirms that defining "misinformed" points is highly subjective and difficult even for humans to agree on.
> > > - I strongly agree with the concern raised by Reviewer TFcU regarding the PICO framework. Strictly categorizing an "answerable question" as a reviewer error is problematic. It ignores a common reality: the reviewer might have missed the information because the paper's clarity was poor. Blaming the reviewer for ambiguity in the writing creates a biased standard for review quality.
> > >
> > > Thank you again to the authors for their thoughtful response and diligent experiments. The reviewers' comments were valuable and insightful, and I hope the authors can use these suggestions to further improve the quality of this work. Keep up the good work!

---

> > > > ### Author Response · Authors · 2025-11-26
> > > >
> > > > We appreciate the reviewer's comments and would like to address the reviewer's remaining concerns.
> > > >
> > > > **W1. The moderate inter-annotator agreement empirically confirms that defining "misinformed" points is highly subjective and difficult even for humans to agree on.**
> > > >
> > > > A. We would like to emphasize that an inter-annotator agreement (IAA) of α = 0.30–0.43 does not indicate that our rubric is highly subjective. Our detailed response to the reviewer’s original concern is provided in the response to **W2** of the reviewer **GJmX**, but for convenience we offer additional clarification here.
> > > >
> > > > Importantly, the moderate IAA values observed in our study stem primarily from the inherent difficulty of the annotation task rather than from subjectivity in the rubric itself. Annotators must read the entire paper, integrate their own domain knowledge, and evaluate premise/claim factuality or question unanswerability. Due to the nature of the task, annotators must accurately identify, understand, and retain all relevant parts of the paper, and their background knowledge must be at least as precise as that of the authors and reviewers to produce reliable annotations. Any mistake along this chain can create disagreements. And because most labels fall within the 4–5 range, a single annotator’s mistake among the three can substantially lower the IAA score.
> > > > In review evaluation settings, what matters most is not the raw IAA value but the reliability of the final label. Although higher IAA would certainly increase confidence, the intrinsic difficulty of the task naturally pushes IAA downward. This difficulty is also reflected in prior review evaluation studies, which typically employ two main annotators and rely on a third annotator to resolve disagreements (Shin et al., 2025; Purkayastha et al., 2025; Du et al., 2024). In contrast, to ensure reliable labeling for every instance, we deliberately chose a stricter protocol: three independent annotators per instance, and the median of their scores as the final label.
> > > > As reported in Appendix E, annotator groups with stronger domain expertise and higher topical relevance to the papers show significantly higher IAA values. This supports our view that moderate IAA mostly comes from difficulty of the annotation task rather than from the subjectivity in the rubric.
> > > > Finally, in the context of factuality evaluation, an IAA of 0.30–0.43 is not unusually low. For example, Ye et al. (2024) report an IAA of 0.42 for factuality judgments made by three annotators in non-expert domains, and this was regarded as reasonable. Given that our task involves expert-level factuality judgments, we believe that our IAA levels are well within an acceptable and meaningful range.
> > > >
> > > > **W2. Strictly categorizing an "answerable question" as a reviewer error is problematic. It ignores a common reality: the reviewer might have missed the information because the paper's clarity was poor. Blaming the reviewer for ambiguity in the writing creates a biased standard for review quality.**
> > > >
> > > > A. In short, if a paper’s clarity is poor so that the relevant information is not stated explicitly in the paper, then the reviewer’s question becomes an "un"answerable question by definition. Therefore, this scenario is not a concern for our framework. Our full response to the reviewer’s original concern is provided in the response to **W1** of the reviewer **TFcU**, which is as follows.
> > > >
> > > > > If a paper is poorly written and especially if a paper's clarity is misleading as the reviewer mentions, then there are less chance to clearly answer the reviewer's question by the paper. In this case, as opposed to what the reviewer argues, most of clarification questions cannot be answered by the paper and are desirable to ask to clarify the paper's writing.
> > > >
> > > > ---
> > > > [1] Mind the Blind Spots: A Focus-Level Evaluation Framework for LLM Reviews. Shin et al., 2025.
> > > > [2] LazyReview A Dataset for Uncovering Lazy Thinking in NLP Peer Reviews. Purkayastha et al., 2025.
> > > > [3] LLMs assist NLP researchers: Critique paper (meta-) reviewing. Du et al., 2024.
> > > > [4] Flask: Fine-grained language model evaluation based on alignment skill sets. Ye et al., 2024.

---

### Official Review · Reviewer_ht6d · 2025-10-30

**Soundness:** 2
**Presentation:** 2
**Contribution:** 2
**Rating:** 4
**Confidence:** 3

**Summary:**

This paper introduces REVIEWSCORE, a framework to automatically detect misinformed review points in peer reviews from AI conferences. A misinformed review point is defined as either (1) a question already answered by the paper or (2) a weakness based on incorrect or unsupported claims. The authors develop two scoring systems: BASE REVIEWSCORE and ADVANCED REVIEWSCORE

The authors construct a human-annotated dataset of review points from 40 ICLR 2021–2023 papers, labeled by 15 graduate students. They then evaluate agreement between these human annotations and predictions from 8 SOTA LLMs (GPT-4, Claude 2, PaLM 2, Gemini, etc.), finding that ADVANCED REVIEWSCORE achieves higher alignment than BASE, especially in weakness detection.

**Strengths:**

* Well-defined problem: The concept of “misinformed review points” is clearly formalized and reflects real concerns authors face, such as reviewers asking questions that are already answered in the paper.
* Argument reconstruction engine: The paper presents a novel approach that extracts both explicit and implicit premises from weaknesses, enabling premise-level factuality evaluation—a step beyond most prior work.
* Empirical study with LLMs: Eight LLMs are evaluated on the REVIEWSCORE dataset, with quantitative results showing moderate human-model agreement, and ADVANCED scoring improves over BASE.

**Weaknesses:**

* Insufficient dataset size: The dataset only covers 40 ICLR submissions across three years (~0.4% of the total), which is far too small to support general conclusions or train reliable automated systems. Most findings should be considered preliminary.
* Limited diversity and experience of annotators: All annotations are from graduate students. There’s no evidence that experienced reviewers, ACs, or SACs were consulted—this may bias the labeling toward an author-centric view of review quality.
* No practical deployment pathway: The paper does not explore how REVIEWSCORE could be integrated into real conference workflows (e.g., as part of rebuttal, AC dashboards, or review filtering), nor does it simulate such use cases.

**Questions:**

* How well does your argument reconstruction engine perform on an external dataset or human-reconstructed arguments?
* Have you tested REVIEWSCORE on reviews from other venues to see how it generalizes beyond the training data?
* Could REVIEWSCORE be used as part of a rebuttal assistant or reviewer feedback system? If yes, what would that pipeline look like?

---

> ### Author Response · Authors · 2025-11-21
> **Rebuttal by Authors (1/3)**
>
> We thank the reviewer for the constructive and detailed comments. **Q** denotes a Question, and **W** denotes a Weakness.
>
> **Misunderstanding in Summary**
>
> We observe that there are slight misunderstandings on the reviewer's summary in the following:
>
> > They then evaluate agreement between these human annotations and predictions from 8 SOTA LLMs (GPT-4, Claude 2, PaLM 2, Gemini, etc.), finding that ADVANCED REVIEWSCORE achieves higher alignment than BASE, especially in weakness detection.
>
> To clarify our experiments, we used five proprietary models (claude-sonnet-3.7, claude-sonnet-4, gpt-4o, gpt-5, and gemini-2.5-flash) and three open-sourced models (qwen3-235b-a22b, deepseek-v3, and llama-3.3). We do not used gpt-4, claude 2, and PaLM 2 to evaluate LLM's ability to evaluate ReviewScore.
> Furthermore, to clarify our method, Base ReviewScore and Advanced ReviewScore share the same scoring criteria for questions and weaknesses that are claims. The only difference is how those evaluate weaknesses that are arguments. Therefore, "ADVANCED REVIEWSCORE achieves higher alignment than BASE, especially in weakness detection" contains a slight misunderstanding of our method.
>
> **W1. The dataset only covers 40 ICLR submissions across three years, which is too small to support general conclusions or train reliable automated systems.**
>
> A. We would like to emphasize that large-scale data collection is fundamentally challenging because human experts must label both the review points and all of their underlying premises. We also clarify that our proposed method is designed to *evaluate* a model’s ability to predict ReviewScore; therefore, no model training is required for our approach.
>
> Compared to other works on review evaluation, we want to highlight that evaluating LLMs on reviews of 40 ICLR submissions is a plausible choice. A detailed comparison is shown in the table below.
>
> | Dataset                       | # Papers | # Review Points | # Premises | # Annotators / Instance | # Total Annotators | Domains              | Venues                 |
> | ------------------            | -------- | --------------- | ---------- | ----------------------- | ------------------ | -------------------- | ---------------------- |
> | **Ou et al. (2025)**          | 41       | 168             | N/A        | –                       | 6                  | NLP                  | NeurIPS 2023–2024      |
> | **Shin et al. (2025)**        | 68       | 327             | N/A        | 2                       | 2                  | –                    | ICLR 2021–2024         |
> | **Purkayastha et al. (2025)** | –        | 500             | N/A        | 2                       | 4+                 | NLP                  | ARR 2022               |
> | **Du et al. (2024)**          | 100      | N/A             | N/A        | 2                       | 40                 | NLP                  | ICLR/NeurIPS 2020–2023 |
> | **Our's**                     | 40       | **657**         | **1,748**  | 3                       | 15                 | CV, NLP, Time Series | ICLR 2021–2023         |
>
> In the table, "-" indicates the information is not explicitly addressed in the paper. We indicate the statistics of human-annotated reviews.
> In terms of number of papers, our work has a similar size with Ou et al. (2025) and has a smaller size than Shin et al. (2025) and Du et al. (2024).
> However, in terms of number of review points, which is the primary measure of review data size, our work has a larger data size than Ou et al. (2025), Shin et al. (2025), and Purkayastha et al. (2025) since they filter out reviews that does not fit to their problem setting (We use all review points as targets for evaluation). Furthermore, while other works focus on evaluating at a level of review points, we also evaluate at a level of premises of review points, which are more fine-grained. As a result, our human experts annotate total 1,748 premises of review points.
> To guarantee annotation reliability, each instance is labeled by three human experts. This design choice intentionally prioritizes quality over scale, resulting in a smaller dataset under limited human annotation capacity.
> We also want to emphasize that other works mostly evaluate reviews of NLP papers, whereas our work includes other domains such as CV and Time Series.
> We add this comparison in Appendix I.1.
>
> In addition, we want to highlight that the purpose of building a human-annotated dataset is to *evaluate* model's ability to evaluate ReviewScore, not to *train* a model or an automated system.
> Therefore, we argue that the review data size is sufficiently large and thus reliable to evaluate ReviewScore evaluation.

---

> > ### Author Response · Authors · 2025-11-21
> > **Rebuttal by Authors (2/3)**
> >
> > **W2. All annotations are from graduate students. There’s no evidence that experienced reviewers, ACs, or SACs were consulted—this may bias the labeling toward an author-centric view of review quality.**
> >
> > A. We want to emphasize that annotators are not labeling reviews as “good” or “bad” in a holistic or subjective sense. Instead, they label only two narrowly defined, fine-grained criteria. As long as annotators sufficiently understand the paper itself, they can provide objective and consistent labels regardless of their prior peer-review experience.
> > To be specific, annotators have to evaluate factuality or unanswerability of review points, which are highly objective and straightforward criteria. If the criteria were value-laden or not specific, then the annotators' experience on the peer review system would matter, but in this case, since the criteria are evidence-based and objective, the only required skills for annotators are to fully understand the paper and have a relevant background knowledge. We think that graduate students suffice these requirements.
> >
> > In addition, other works on review evaluation (Ou et al., 2025; Shin et al., 2025; Purkayastha et al., 2025) mostly use graduate students as human annotators.
> > Despite these, we acknowledge that there could be unavoidable noise in human annotation, so we left this as a limitation, which is described in Section 5 line 467-471. We also further analyzed human annotations in Appendix E, specifically their expertise and inter-annotator agreement for each annotator group.
> >
> > **W3. The paper does not explore how REVIEWSCORE could be integrated into real conference workflows (e.g., as part of rebuttal, AC dashboards, or review filtering), nor does it simulate such use cases.**
> >
> > A. In Appendix A, we summarize how automatic ReviewScore evaluation could benefit different roles of the current peer reviewing system. Detailed explanations are provided in the response of **Q3**.
> >
> > For the second concern, we agree that it is important to simulate how ReviewScore evaluation could benefit the current peer review dynamics. However, human simulation costs far beyond our budget and capability, and replacing all humans as AI agents are not fully reliable and also requires huge API costs that goes beyond our budget, so we leave the simulation as a future work.
> >
> > **Q1. How well does your argument reconstruction engine perform on an external dataset or human-reconstructed arguments?**
> >
> > A. This work is targeting paper reviews which include scientific arguments, and as the reviewer mentions, we are trying to generalize the proposed argument reconstruction engine into other domains such as political/societal arguments for our next work. Since these two arguments use different types of reasoning, especially political/societal arguments include lots of inductive reasoning and sometimes contain logical fallacies, we think that the engine should be revised accordingly. However, we think that this discussion is out of scope of this work.

---

> > > ### Author Response · Authors · 2025-11-21
> > > **Rebuttal by Authors (3/3)**
> > >
> > > **Q2. Have you tested REVIEWSCORE on reviews from other venues to see how it generalizes beyond the training data?**
> > >
> > > A. Before answering the question, we want to make sure that our method does not require any kind of model training, so we do not need any training data. If the reviewer is asking whether evaluation LLMs generalize to papers and reviews submitted after the model’s knowledge cutoff, we were unable to identify an appropriate venue for such testing. Specifically, we could not find a major AI conference—whose proceedings occur after the knowledge cutoff of current state-of-the-art models—that publicly releases both submission drafts and their corresponding reviews. At this point, ICLR 2026 perfectly satisfies all of these criteria, but to test generalizability of models, we have to additionally hire human experts to annotate reviews, which requires significant amount of time and costs, so we are leaving it as a future work due to the limited discussion period.
> > >
> > > If the reviewer is asking whether a ReviewScore evaluation would generalize on other AI conferences, we perform an extra experiment on reviews from ARR 2022. Due to the limited discussion period, we use a human expert to annotate reviews of a single paper in the NLPeer dataset (Dycke et al., 2023). Following the same setup with our main experiment, claude-sonnet-3.7 achieves 0.385 Kappa score and 0.333 F1 score, which shows similar human-model alignment with our main result. This result shows a potential of ReviewScore to generalize on other AI conferences.
> > > Furthermore, other works on review evaluation (Ou et al., 2025; Shin et al., 2025; Purkayastha et al., 2025) also target a single AI conference. However, as shown in the table in the response of **W1**, we covered not only NLP domain but also other domains such as CV and Time Series.
> > > We also want to point out that outside of AI conferences is out of our motivation. Misinformed reviews are problems of current AI conferences, not in other domains, so we want to clarify that these other domains are not our target. However, since ReviewScore is a domain-generalizable evaluation framework, we think that it would work on other academic domains. We leave testing this generalizability as a future work.
> > >
> > > **Q3. Could REVIEWSCORE be used as part of a rebuttal assistant or reviewer feedback system? If yes, what would that pipeline look like?**
> > >
> > > A. Yes, it could be used both as a rebuttal assistant for authors and as a feedback system for reviewers.
> > > As we mentioned in the response of **W3**, we discussed this point in Appendix A as follows.
> > >
> > > > For authors, by providing reconstruction of argumentative review points, it helps to understand or clarify the reviewer's intention and to formulate a rebuttal.
> > > For reviewers, by providing ReviewScore of their review points, it allows them to verify the review quality by themselves and helps reviewers to better understand the paper.
> > > For metareviewers, by providing ReviewScore of each reviewer, it could assist their final decisions.
> > >
> > > To clarify from the point of reviewers, after reviewers leave their comments, we could run the automatic ReviewScore evaluation to check whether there are any incorrect premises or could be already answered by the paper. This feedback is sent to reviewers, then reviewers could correct their misunderstandings or resolve their questions. This feedback loop makes reviewers revise their comments and ultimately lead to more helpful and higher-quality reviews than before.
> > >
> > > ---
> > > [1] CLAIMCHECK: How Grounded are LLM Critiques of Scientific Papers? Ou et al., 2025.
> > > [2] Mind the Blind Spots: A Focus-Level Evaluation Framework for LLM Reviews. Shin et al., 2025.
> > > [3] LazyReview A Dataset for Uncovering Lazy Thinking in NLP Peer Reviews. Purkayastha et al., 2025.
> > > [4] LLMs assist NLP researchers: Critique paper (meta-) reviewing. Du et al., 2024.
> > > [5] NLPeer: A unified resource for the computational study of peer review. Dycke et al., 2023.

---

> ### Author Response · Authors · 2025-11-27
> **Reminder for Paper Discussion**
>
> Dear Reviewer ht6d,
>
> We appreciate the time and effort you have dedicated to reviewing our paper. We hope our responses have addressed your concerns. If you have any further questions or suggestions, we would be grateful to hear them. Thank you once again for your valuable feedback throughout this process!
>
> Best,
> Authors

---

> ### Comment · Reviewer_ht6d · 2025-11-27
>
> Thank you for the detailed response. Most of my concerns have been addressed. My primary remaining concern relates to the scale of the dataset. I fully understand the challenges and substantial human effort required to curate a dataset of this nature—such efforts are rare in our community. However, I would encourage the authors to consider increasing the dataset size to strengthen the validation, especially given that ICLR submissions in recent years have exceeded 10k.

---

> > ### Author Response · Authors · 2025-11-28
> >
> > We appreciate the reviewer's comments and would like to address the reviewer's remaining concern.
> >
> > > My primary remaining concern relates to the scale of the dataset. I would encourage the authors to consider increasing the dataset size to strengthen the validation, especially given that ICLR submissions in recent years have exceeded 10k.
> >
> > A. We would like to emphasize that the size of our dataset is fully consistent with academic practices and reflects a quality-oriented design. In the expert domain, we annotated a total of 2,405 instances, including both review points and their associated premises. Within the review-evaluation literature, Shin et al. (2025) annotated 327 instances, Purkayastha et al. (2025) annotated 500 instances, and Sadallah et al. (2025) annotated 1,430 instances. Similarly, in the LLM-evaluation literature, Ye et al. (2024) annotated 1,740 instances, Han et al. (2025) annotated 1,994 instances, and An et al. (2023) annotated over 2,000 instances. Taken together, these numbers indicate that the scale of our human annotation lies well within the range commonly accepted in the review-evaluation and LLM-evaluation communities, and is sufficient for validating our hypotheses. They also illustrate that a trade-off between data scale and annotation quality is standard in this line of work.
> >
> > ---
> > [1] Mind the Blind Spots: A Focus-Level Evaluation Framework for LLM Reviews. Shin et al., 2025.
> > [2] LazyReview A Dataset for Uncovering Lazy Thinking in NLP Peer Reviews. Purkayastha et al., 2025.
> > [3] The Good, the Bad and the Constructive: Automatically Measuring Peer Review's Utility for Authors. Sadallah et al., 2025.
> > [4] Flask: Fine-grained language model evaluation based on alignment skill sets. Ye et al., 2024.
> > [5] Judge's Verdict: A Comprehensive Analysis of LLM Judge Capability Through Human Agreement. Han et al., 2025.
> > [6] L-Eval: Instituting Standardized Evaluation for Long Context Language Models. An et al., 2023.

---

### Official Review · Reviewer_TFcU · 2025-10-31

**Soundness:** 2
**Presentation:** 3
**Contribution:** 1
**Rating:** 2
**Confidence:** 3

**Summary:**

This paper attempts to detect low-quality reviews by defining a misinformed review point, i.e., ReviewScore. Specifically, the misinformed review point includes a question stated in a review that can already be answered by the paper, or a weakness stated in a review is incorrect or contains incorrect premises regarding the paper. After that, the authors build a human expert-annotated benchmark to evaluate the ability of current SOTA LLMs.

**Strengths:**

1. The problem studied in this paper is valuable.
2. The paper is well-organized.

**Weaknesses:**

1. My main concern is the proposed definition 2 (Misinformed Review Point), which is the core assumption of this paper. The claim "a question stated in a review can already be answered by the paper" may be caused by many other cases, not just the misinformed point. For example, maybe the paper is poorly written, the clarity is misleading, or the reviewer is not familiar with this area, and so on. If this paper is poorly written
2. We believe that, in the peer-review system, scholars evaluate their academic level rankings based on the "consistency assumption", i.e., scholars with stronger abilities usually have stronger persuasiveness for evaluating others, and these scholars can also obtain higher achievements [1]. Therefore, we believe the "review quality" is mainly determined by the human/expert preference. And the "low-quality review problem" is a system problem.


[1] PICO: PEER REVIEW IN LLMS BASED ON CONSISTENCY OPTIMIZATION. ICLR'2025.

**Questions:**

See the weaknesses.

---

> ### Author Response · Authors · 2025-11-21
> **Rebuttal by Authors**
>
> We thank the reviewer for the constructive and detailed comments. **W** denotes a Weakness.
>
> **W1. The claim "a question stated in a review can already be answered by the paper" may be caused by many other cases (e.g., the paper is poorly written, the clarity is misleading, or the reviewer is not familiar with this area), not just the misinformed point.**
>
> A. If a paper is poorly written and especially if a paper's clarity is misleading as the reviewer mentions, then there are less chance to clearly answer the reviewer's question by the paper. In this case, as opposed to what the reviewer argues, most of clarification questions cannot be answered by the paper and are desirable to ask to clarify the paper's writing.
> If a reviewer is not familiar with the area so that its reviews are misinformed, then it should also be flagged as misinformed review points. We want to emphasize that although the reviewer is properly matched with the paper, if the reviewer does not carefully read the paper, then misinformed reviews could also be submitted. Instead of simply blaming the peer review system, our goal is to use ReviewScore to detect misinformed reviews, report those back to reviewers to make them re-write before they submit reviews or send those to meta-reviewers to assist their final decisions, so that we could shift the peer review dynamics in a more constructive way.
>
> **W2. We believe the "review quality" is mainly determined by the human/expert preference, and the "low-quality review problem" is a system problem.**
>
> A. We want to emphasize that our contribution is complementary and orthogonal to measuring review quality by human expert preferences. We agree that it is important to evaluate review quality based on experienced human experts' knowledge and thoughts, but at the same time, we want to highlight that it is also important to detect low-quality reviews by defining their failure modes in order to resolve the current issue of the peer review system.
> To clearly understand this low-quality review issue, our first contribution is that we suggest trustworthy criteria to decide whether a review is low-quality or not, and our second contribution is that based on those criteria, we build an automated scoring system to detect misinformed reviews.
> In addition, there are other concurrent works (Goldberg et al., 2025; Guo et al., 2023; Sadallah et al., 2025; Ou et al., 2025; Purkayastha et al., 2025; Du et al., 2024) that study review quality evaluation in different ways, which supports the importance of the problem and interest of the NLP community.
>
> ---
> [1] Peer reviews of peer reviews: A randomized controlled trial and other experiments. Goldberg et al., 2025.
> [2] Automatic analysis of substantiation in scientific peer reviews. Guo et al., 2023.
> [3] The Good, the Bad and the Constructive: Automatically Measuring Peer Review's Utility for Authors. Sadallah et al., 2025.
> [4] CLAIMCHECK: How Grounded are LLM Critiques of Scientific Papers? Ou et al., 2025.
> [5] LazyReview A Dataset for Uncovering Lazy Thinking in NLP Peer Reviews. Purkayastha et al., 2025.
> [6] LLMs assist NLP researchers: Critique paper (meta-) reviewing. Du et al., 2024.

---

> ### Author Response · Authors · 2025-11-27
> **Reminder for Paper Discussion**
>
> Dear Reviewer TFcU,
>
> We appreciate the time and effort you have dedicated to reviewing our paper. We hope our responses have addressed your concerns. If you have any further questions or suggestions, we would be grateful to hear them. Thank you once again for your valuable feedback throughout this process!
>
> Best,
> Authors

---

### Official Review · Reviewer_GJmX · 2025-11-06

**Soundness:** 3
**Presentation:** 3
**Contribution:** 3
**Rating:** 8
**Confidence:** 3

**Summary:**

The paper introduces ReviewScore, a novel framework and dataset for detecting misinformed peer review points in academic paper reviews. They are defined as (1) questions already answered in the paper or (2) weaknesses based on incorrect premises.

The authors verify that a significant fraction of review points (15.2% of weaknesses, 26.4% of questions) are misinformed.

To automate this detection, they proposes a two-stage evaluation approach:
1. Base ReviewScore, measuring unanswerability and factuality on a 5-point scale.
2. Advanced ReviewScoreE, which decomposes review points into arguments and reconstructs their premises via an automatic argument reconstruction engine, checked for validity (using a SAT solver) and faithfulness (via LLM feedback loops).

A human-annotated dataset of 657 review points (ICLR 2021–2023) is created to benchmark human–model agreement across eight modern LLMs (Claude, GPT-4/5, Gemini, LLaMA-3, etc.). Experiments show moderate agreement (F1≈0.4–0.5, κ≈0.3–0.4), and the paper demonstrates that premise-level analysis significantly improves over direct weakness-level evaluation.

**Strengths:**

1. Novel Problem Definition: the paper identifies misinformed reviews as a measurable and impactful aspect of review quality — a previously underexplored issue. The operational definitions (answered question / incorrect premise) are specific yet widely applicable.

2. Methodological Innovation: The automatic argument reconstruction engine combining LLMs, formal logic translation, and a SAT solver is a creative and rigorous integration of symbolic reasoning and LLM capabilities. The validity–faithfulness feedback loops are well-motivated and thoughtfully designed.

3. High-quality Dataset: The authors build a trustworthy expert-annotated dataset (657 review points, 1,748 premises) with strong documentation and process controls (cross-annotation, consensus, training sessions).

**Weaknesses:**

1. Limited Agreement and Practical Readiness: Despite the substantial improvement over the baseline, the overall human-model agreement remains moderate (F1 ~0.45, Kappa ~0.35). This level of reliability is likely insufficient for fully autonomous deployment and would require human oversight, potentially reducing the net efficiency gains.

2. Subjectivity in Annotations: Inter-annotator α=0.30–0.43 is low, suggesting high ambiguity in human judgments of factuality. The paper’s conclusions about “moderate agreement” might partly reflect human inconsistency rather than LLM capability.

3. Dataset Scope and Diversity: Only ICLR reviews are used; the framework’s generality to NLP, CVPR, or smaller venues is unclear. Annotators are graduate students rather than senior researchers — potentially limiting domain depth.

4. Evaluation Simplifications Models were provided only the main paper text, not figures or appendices, which could substantially affect factual verification. The binary misinformed/not-misinformed threshold is somewhat arbitrary; alternative calibration curves could provide richer insight.

5. Argument Reconstruction Validation: While impressive, the argument reconstruction quality is mainly assessed via one model (Claude 3.7). Broader ablations (e.g., GPT-5 vs open models) or human inspection metrics could strengthen confidence.

**Questions:**

1. How consistent are human labels for the same review points across groups (e.g., claim vs argument disagreements)?
2. Could the reconstruction engine generalize to other reasoning tasks (e.g., debate analysis, scientific claim verification)?
3. Could incorporating citation graphs or author responses systematically (rather than as an optional experiment) raise reliability?

**Details Of Ethics Concerns:**

No ethics issue detected.

---

> ### Author Response · Authors · 2025-11-21
> **Rebuttal by Authors (1/3)**
>
> We thank the reviewer for the constructive and detailed comments. **Q** denotes a Question, and **W** denotes a Weakness.
>
> **W1. The overall human-model agreement remains moderate (F1 ~0.45, Kappa ~0.35). This level of reliability is likely insufficient for fully autonomous deployment and would require human oversight.**
>
> A. First, to address this concern, we would like to refer a relevant part in the Limitations section.
>
> > Throughout Section 4, human-model agreement seems low even with authors response.
> However, compared to Du et al. (2024) which measures human-model agreement on *Deficient* review detection and reports the best F1 score as 0.220, our work reports the best F1 score as 0.449, indicating more than $2 \times$ agreement.
>
> This supports that our criteria or the way of measuring those are clearer and more objective than the previous work.
> Second, since there are some cases where human labels diverge, if we change the way to measure human-model agreement to whether a model correctly predicts at least one human label, then the agreement score would increase.
> Despite these, as the reviewer mentions, to fully automate ReviewScore evaluation, we agree that evaluation models should be more advanced and should have higher level of human alignment.
>
> **W2. Inter-annotator α=0.30–0.43 is low, suggesting high ambiguity in human judgments of factuality. The paper’s conclusions about “moderate agreement” might partly reflect human inconsistency rather than LLM capability.**
>
> A. We want to clarify that the reason of low inter-annotator agreement is from some incorrect human judgments, not from ambiguous evaluation criteria. Also, Appendix E.2, we confirm that high-expertise or high-relevance in human annotation alleviates this issue. To explain this with more details, we would like to refer a relevant part in Appendix E.2.
>
> > By comparing inter-group agreements, Group 2 and 4 show significantly higher ReviewScore agreement than other groups.
> In contrast, Group 1 and 5 include negative inter-annotator agreement, which is a critical concern for reliable human annotation. By a manual disagreement analysis, we confirm that most disagreements come from human annotators with low paper relevance (i.e., anno\_11 and anno\_51 in Table 8).
> To summarize, we observe that the inter-annotator agreement aligns with the annotator expertise (Table 8), and we (unsurprisingly) confirm that human annotations could be more reliable if their research interests become more relevant to the assigned papers.
>
> In addition, compared to other works that measure inter-annotator agreement on factuality, α=0.30–0.43 is a valid level of agreement. In Ye et al. (2024), they measured inter-annotator agreement of factuality mostly on non-expertise domains, where the annotation would be easier than our's, and their reported α was 0.42.

---

> > ### Author Response · Authors · 2025-11-21
> > **Rebuttal by Authors (2/3)**
> >
> > **W3. Only ICLR reviews are used; the framework’s generality to NLP, CVPR, or smaller venues is unclear. Annotators are graduate students rather than senior researchers — potentially limiting domain depth.**
> >
> > A. To address the first concern, we include the response to **Q2** of the reviewer **ht6d** in the following.
> >
> > > If the reviewer is asking whether a ReviewScore evaluation would generalize on other AI conferences, we perform an extra experiment on reviews from ARR 2022. Due to the limited discussion period, we use a human expert to annotate reviews of a single paper in the NLPeer dataset (Dycke et al., 2023). Following the same setup with our main experiment, claude-sonnet-3.7 achieves 0.385 Kappa score and 0.333 F1 score, which shows similar human-model alignment with our main result. This result shows a potential of ReviewScore to generalize on other AI conferences.
> > Furthermore, other works on review evaluation (Ou et al., 2025; Shin et al., 2025; Purkayastha et al., 2025) also target a single AI conference. However, as shown in the table in the response of **W1**, we covered not only NLP domain but also other domains such as CV and Time Series.
> >
> > To address the second concern, we include the response to **W2** of the reviewer **ht6d** in the following.
> >
> > > We want to emphasize that annotators are not labeling reviews as “good” or “bad” in a holistic or subjective sense. Instead, they label only two narrowly defined, fine-grained criteria. As long as annotators sufficiently understand the paper itself, they can provide objective and consistent labels regardless of their prior peer-review experience.
> > To be specific, annotators have to evaluate factuality or unanswerability of review points, which are highly objective and straightforward criteria. If the criteria were value-laden or not specific, then the annotators' experience on the peer review system would matter, but in this case, since the criteria are evidence-based and objective, the only required skills for annotators are to fully understand the paper and have a relevant background knowledge. We think that graduate students suffice these requirements.
> > >
> > > In addition, other works on review evaluation (Ou et al., 2025; Shin et al., 2025; Purkayastha et al., 2025) mostly use graduate students as human annotators.
> > Despite these, we acknowledge that there could be unavoidable noise in human annotation, so we left this as a limitation, which is described in Section 5 line 467-471. We also further analyzed human annotations in Appendix E, specifically their expertise and inter-annotator agreement for each annotator group.
> >
> > **W4. Models were provided only the main paper text, not figures or appendices, which could substantially affect factual verification. The binary misinformed/not-misinformed threshold is somewhat arbitrary; alternative calibration curves could provide richer insight.**
> >
> > A. For the first concern, we agree with the reviewer. However, in our dataset, there are very few cases where figures or appendices are necessary for factual verification, and in terms of practice, including all figures and appendices increases the number of input tokens by several times which is out of our budget, so we decided not to include those. But theoretically we could evaluate ReviewScore by considering both. The reviewer could find this limitation in line 461-463.
> >
> > For the second concern, as we mention in line 156 that points to Figure 15 and 16 in Appendix H, Score 1-2 correspond to false premise or answerable question by the paper, Score 4-5 correspond to true premise or unanswerable question by the paper, and Score 3 corresponds to no verdict. According to the definition of misinformed review points, Score 1-2 correspond to a misinformed class, and the other scores correspond to a not-misinformed class. Therefore, considering the meaning of rubrics, we do not need further calibrations.
> >
> > **W5. The argument reconstruction quality is mainly assessed via one model (Claude 3.7). Broader ablations (e.g., GPT-5 vs open models) or human inspection metrics could strengthen confidence.**
> >
> > A. To clarify, we do not use claude-sonnet-3.7 to evaluate argument reconstruction quality. The model is used only within the reconstruction engine itself. For evaluation, we use a SAT solver to assess validity, and human annotations to assess faithfulness. The only stage where the model is involved in the evaluation pipeline is in converting natural language into formal language for validity checking; however, because this step produced virtually no errors in practice, we did not conduct additional human evaluation for that part.

---

> > > ### Author Response · Authors · 2025-11-21
> > > **Rebuttal by Authors (3/3)**
> > >
> > > **Q1. How consistent are human labels for the same review points across groups (e.g., claim vs argument disagreements)?**
> > >
> > > A. As we mention in Appendix E.2 and Table 9, claims show higher inter-annotator agreement than arguments. Across annotator groups, the highest agreement was 0.400 for claims and 0.361 for arguments. Questions show similar agreement to claims. For arguments, premise-level factualities (range from 1-5) are aggregated into a single score, whereas for claims and questions, those are directly evaluated as a score from 1-5. We think that the way of measuring scores makes a discrepancy in inter-annotator agreements for different review point types. Detailed results are shown in Appendix E.2.
> > >
> > > **Q2. Could the reconstruction engine generalize to other reasoning tasks (e.g., debate analysis, scientific claim verification)?**
> > >
> > > A. We include the response to **Q1** of the reviewer **ht6d** in the following.
> > >
> > > > This work is targeting paper reviews which include scientific arguments, and as the reviewer mentions, we are trying to generalize the proposed argument reconstruction engine into other domains such as political/societal arguments for our next work. Since these two arguments use different types of reasoning, especially political/societal arguments include lots of inductive reasoning and sometimes contain logical fallacies, we think that the engine should be revised accordingly. However, we think that this discussion is out of scope of this work.
> > >
> > > **Q3. Could incorporating citation graphs or author responses systematically (rather than as an optional experiment) raise reliability?**
> > >
> > > A. For using citation graphs, if a knowledge base to decide premise factuality is "referred papers", it would be more reliable to go through relevant papers using citation graphs and then decide the factuality. However, in our collected dataset, there are rare cases where knowledge bases are referred papers, so we do not include an advanced factuality verification process and simply assume that the evaluation models already have relevant knowledge. We will leave this as a future work.
> > >
> > > For using authors response, as analyzed in Section 4.4, we confirm that authors response benefits detecting misinformed reviews by providing additional cues for model judgments. However, we are thinking of a primary usage of ReviewScore evaluation as a feedback agent to reviewers before they submit their reviews. In this scenario, an authors response is not given yet, so we cannot use it for ReviewScore evaluation, which we leave it as a supplementary experiment.
> > >
> > > ---
> > > [1] LLMs assist NLP researchers: Critique paper (meta-) reviewing. Du et al., 2024.
> > > [2] Flask: Fine-grained language model evaluation based on alignment skill sets. Ye et al., 2024.
> > > [3] NLPeer: A unified resource for the computational study of peer review. Dycke et al., 2023.
> > > [4] CLAIMCHECK: How Grounded are LLM Critiques of Scientific Papers? Ou et al., 2025.
> > > [5] Mind the Blind Spots: A Focus-Level Evaluation Framework for LLM Reviews. Shin et al., 2025.
> > > [6] LazyReview A Dataset for Uncovering Lazy Thinking in NLP Peer Reviews. Purkayastha et al., 2025.

---

### Meta-Review · Area_Chair_REQc · 2025-12-28

**Summary:**

The paper introduces ReviewScore, a framework for detecting misinformed peer review points (e.g., weaknesses based on incorrect premises or questions already answered in the paper) using LLMs and neurosymbolic methods.

While the problem definition is novel and the creation of an expert-annotated dataset is a solid contribution, the work suffers from low inter-annotator agreement, indicating high subjectivity in the definition of "misinformed" points. Additional limitations include the restricted dataset scope (drawn from only 40 ICLR papers), reliance on labels from graduate-student annotators rather than senior researchers, and an overly simplistic binary classification of review points. Though partially addressed, but these issues collectively challenge the framework's generalizability and overall robustness. The work requires further improvement to address these shortcomings. Thus, I recommend rejecting this paper.

**Reviewer Concerns:**

Concerns such as subjectivity in annotations/definitions, limited generalizability, and low inter-annotator agreement remain unaddressed.

**Reviewer Scores:**

No reviewer would change the score.

---

### Decision · Program_Chairs · 2026-01-26

Reject